# D4RL: Datasets for Deep Data-Driven Reinforcement Learning

## Abstract

The offline reinforcement learning (RL) setting (also known as full batch RL), where a policy is learned from a static dataset, is compelling as progress enables RL methods to take advantage of large, previously-collected datasets, much like how the rise of large datasets has fueled results in supervised learning. However, existing *online* RL benchmarks are not tailored towards the *offline* setting and existing offline RL benchmarks are restricted to data generated by partially-trained agents, making progress in offline RL difficult to measure. In this work, we introduce benchmarks specifically designed for the offline setting, guided by key properties of datasets relevant to real-world applications of offline RL. With a focus on dataset collection, examples of such properties include: datasets generated via hand-designed controllers and human demonstrators, multitask datasets where an agent performs different tasks in the same environment, and datasets collected with mixtures of policies. By moving beyond simple benchmark tasks and data collected by partially-trained RL agents, we reveal important and unappreciated deficiencies of existing algorithms. To facilitate research, we have released our benchmark tasks and datasets with a comprehensive evaluation of existing algorithms, an evaluation protocol, and open-source examples. This serves as a common starting point for the community to identify shortcomings in existing offline RL methods and a collaborative route for progress in this emerging area.

## 1 Introduction

Impressive progress across a range of machine learning applications has been driven by high-capacity neural network models with large, diverse training datasets (Goodfellow et al., 2016). While reinforcement learning (RL) algorithms have also benefited from deep learning (Mnih et al., 2015), active data collection is typically required for these algorithms to succeed, limiting the extent to which large, previously-collected datasets can be leveraged. Offline RL (Lange et al., 2012) (also known as *full batch* RL), where agents learn from previously-collected datasets, provides a bridge between RL and supervised learning. The promise of offline RL is leveraging large, previously-collected datasets in the context of sequential decision making, where reward-driven learning can produce policies that reason over temporally extended horizons. This

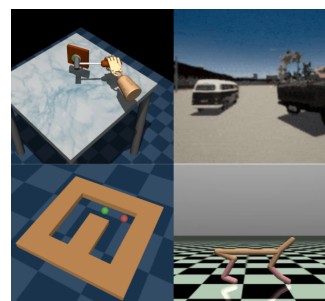

Figure 1: A selection of proposed benchmark tasks.

could have profound implications for a range of application domains, such as robotics, autonomous driving, and healthcare.

Current offline RL methods have not fulfilled this promise yet. While recent work has investigated technical reasons for this (Fujimoto et al., 2018a; Kumar et al., 2019; Wu et al., 2019), a major challenge in addressing these issues has been the lack of standard evaluation benchmarks. Ideally, such a benchmark should: **a)** be composed of tasks that reflect challenges in real-world applications of data-driven RL, **b)** be widely accessible for researchers and define clear evaluation protocols for

---

Website with code, examples, tasks, and data is available at `https://sites.google.com/view/d4rl-anonymous/`

reproducibility, and **c)** contain a range of difficulty to differentiate between algorithms, especially challenges particular to the offline RL setting.

Most recent works (Fujimoto et al., 2018b; Wu et al., 2019; Kumar et al., 2019; Peng et al., 2019; Agarwal et al., 2019b) use existing online RL benchmark domains and data collected from training runs of online RL methods. However, these benchmarks were not designed with offline RL in mind and such datasets do not reflect the heterogenous nature of data collected in practice. Wu et al. (2019) find that existing benchmark datasets are not sufficient to differentiate between simple baseline approaches and recently proposed algorithms. Furthermore, the aforementioned works do not propose a standard evaluation protocol, which makes comparing methods challenging.

**Why simulated environments?** While relying on (existing) real-world datasets is appealing, evaluating a candidate policy is challenging because it weights actions differently than the data collection and may take actions that were not collected. Thus, evaluating a candidate policy requires either collecting additional data from the real-world system, which is hard to standardize and make broadly available, or employing off-policy evaluation, which is not yet reliable enough (e.g., the NeurIPS 2017 Criteo Ad Placement Challenge used off-policy evaluation, however, in spite of an unprecedentedly large dataset because of the variance in the estimator, top entries were not statistically distinguishable from the baseline). Both options are at odds with a widely-accessible and reproducible benchmark. As a compromise, we use high-quality simulations that have been battle-tested in prior domain-specific work, such as in robotics and autonomous driving. These simulators allow researchers to evaluate candidate policies accurately.

Our primary contribution is the introduction of *Datasets for Deep Data-Driven Reinforcement Learning* (D4RL): a suite of tasks and datasets for benchmarking progress in offline RL. We focus our design around tasks and data collection strategies that exercise dimensions of the offline RL problem likely to occur in practical applications, such as partial observability, passively logged data, or human demonstrations. To serve as a reference, we benchmark state-of-the-art offline RL algorithms (Haarnoja et al., 2018b; Kumar et al., 2019; Wu et al., 2019; Agarwal et al., 2019b; Fujimoto et al., 2018a; Nachum et al., 2019; Peng et al., 2019; Kumar et al., 2020) and provide reference implementations as a starting point for future work. While previous studies (e.g., (Wu et al., 2019)) found that all methods including simple baselines performed well on the limited set of tasks used in prior work, we find that most algorithms struggle to perform well on tasks with properties crucial to real-world applications such as passively logged data, narrow data distributions, and limited human demonstrations. By moving beyond simple benchmark tasks and data collected by partially-trained RL agents, we reveal important and unappreciated deficiencies of existing algorithms. To facilitate adoption, we provide an easy-to-use API for tasks, datasets, and a collection of benchmark implementations of existing algorithms (`https://sites.google.com/view/d4rl-anonymous/`). This is a common starting point for the community to identify shortcomings in existing offline RL methods, and provides a meaningful metric for progress in this emerging area.

## 2 RELATED WORK

Recent work in offline RL has primarily used datasets generated by a previously trained behavior policy, ranging from a random initial policy to a near-expert online-trained policy. This approach has been used for continuous control for robotics (Fujimoto et al., 2018a; Kumar et al., 2019; Wu et al., 2019; Gulcehre et al., 2020), navigation (Laroche et al., 2019), industrial control (Hein et al., 2017), and Atari video games (Agarwal et al., 2019b). To standardize the community around common datasets, several recent works have proposed benchmarks for offline RL algorithms. Agarwal et al. (2019b); Fujimoto et al. (2019) propose benchmarking based on the discrete Atari domain. Concurrently to our work, Gulcehre et al. (2020) proposed a benchmark[1] based on locomotion and manipulation tasks with perceptually challenging input and partial observability. While these are important contributions, both benchmarks suffer from the same shortcomings as prior evaluation protocols: they rely on data collected from online RL training runs. In contrast, with D4RL, in addition to collecting data from online RL training runs, we focus on a range of dataset collection

---

[1]Note that the benchmark proposed by Gulcehre et al. (2020) contains the Real-World RL Challenges benchmark (Dulac-Arnold et al., 2020) based on (Dulac-Arnold et al., 2019) and also uses data collected from partially-trained RL agents.

procedures inspired by real-world applications, such as human demonstrations, exploratory agents, and hand-coded controllers. As alluded to by Wu et al. (2019) and as we show in our experiments, the performance of current methods depends strongly on the data collection procedure, demonstrating the importance of modeling realistic data collection procedures in a benchmark.

Offline RL using large, previously-collect datasets has been successfully applied to real-world systems such as in robotics (Cabi et al., 2019), recommender systems (Li et al., 2010; Strehl et al., 2010; Thomas et al., 2017), and dialogue systems (Henderson et al., 2008; Pietquin et al., 2011; Jaques et al., 2019). These successes point to the promise of offline RL, however, they rely on private real-world systems or expensive human labeling for evaluation which is not scalable or accessible for a benchmark. Moreover, significant efforts have been made to incorporate large-scale datasets into off-policy RL (Kalashnikov et al., 2018; Mo et al., 2018; Gu et al., 2017), but these works use large numbers of robots to collect online interaction during training. Pure online systems for real-world learning have included model-based approaches (Hester et al., 2011), or approaches which collect large amounts of data in parallel (Gu et al., 2017). While we believe these are promising directions for future research, the focus of this work is to provide a varied and accessible platform for developing algorithms.

## 3 BACKGROUND

The offline reinforcement learning problem statement is formalized within a Markov decision process (MDP), defined by a tuple $(\mathcal{S}, \mathcal{A}, P, R, \rho_0, \gamma)$, where $\mathcal{S}$ denotes the state space, $\mathcal{A}$ denotes the action space, $P(s'|s, a)$ denotes the transition distribution, $\rho_0(s)$ denotes the initial state distribution, $R(s, a)$ denotes the reward function, and $\gamma \in (0, 1)$ denotes the discount factor. The goal in RL is to find a policy $\pi(a|s)$ that maximizes the expected cumulative discounted rewards $J(\pi) = E_{\pi, P, \rho_0}[\sum_{t=0}^{\infty} \gamma^t R(s_t, a_t)]$, also known as the discounted returns.

In episodic RL, the algorithm is given access to the MDP via trajectory samples for arbitrary $\pi$ of the algorithm's choosing. Off-policy methods may use experience replay (Lin, 1992) to store these trajectories in a *replay buffer* $\mathcal{D}$ of transitions $(s_t, a_t, s_{t+1}, r_t)$, and use an off-policy algorithm such as Q-learning (Watkins & Dayan, 1992) to optimize $\pi$. However, these methods still iteratively collect additional data, and omitting this collection step can produce poor results. For example, running state-of-the-art off-policy RL algorithms on trajectories collected from an expert policy can result in diverging Q-values (Kumar et al., 2019).

In *offline RL*, the algorithm no longer has access to the MDP, and is instead presented with a fixed *dataset* of transitions $\mathcal{D}$. The (unknown) policy that generated this data is referred to as a *behavior policy* $\pi_B$. Effective offline RL algorithms must handle distribution shift, as well as data collected via processes that may not be representable by the chosen policy class. Levine et al. (2020) provide a comprehensive discussion of the problems affecting offline RL.

## 4 TASK DESIGN FACTORS

In order to design a benchmark that provides a meaningful measure of progress towards realistic applications of offline RL, we choose datasets and tasks to cover a range of properties designed to challenge existing RL algorithms. We discuss these properties as follows:

**Narrow and biased data distributions**, such as those from deterministic policies, are problematic for offline RL algorithms and may cause divergence both empirically (Fujimoto et al., 2018a; Kumar et al., 2019) and theoretically (Munos, 2003; Farahmand et al., 2010; Kumar et al., 2019; Agarwal et al., 2019a; Du et al., 2020). Narrow datasets may arise in human demonstrations, or when using hand-crafted policies. An important challenge in offline RL is to be able to gracefully handle diverse data distributions without algorithms diverging or producing performance worse than the provided behavior. A common approach for dealing with such data distributions is to adopt a conservative approach which tries to keep the behavior close to the data distribution (Fujimoto et al., 2018a; Kumar et al., 2019; Wu et al., 2019).

**Undirected and multitask data** naturally arises when data is passively logged, such as recording user interactions on the internet or recording videos of a car for autonomous driving. This data may not necessarily be directed towards the specific task one is trying to accomplish. However,

pieces of trajectories can still provide useful information to learn from. For example, one may be able to combine sub-trajectories to accomplish a task. In the figure to the upper-right, if an agent is given trajectories from A-B and B-C in a dataset (left image), it can form a trajectory from A-C by combining the corresponding halves of the original trajectories. We refer to this property as *stitching*, since the agent can use portions of existing trajectories in order to solve a task, rather than relying on generalization outside of the dataset.

**Sparse rewards**. Sparse reward problems pose challenges to traditional RL methods due to the difficulty of credit assignment and exploration. Because offline RL considers fixed datasets without exploration, sparse reward problems provide an unique opportunity to isolate the ability of algorithms to perform credit assignment decoupled from exploration.

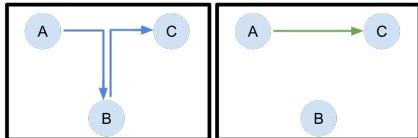

Figure 2: An example of stitching together subtrajectories to solve a task.

**Suboptimal data**. For tasks with a clear objective, the datasets may not contain behaviors from optimal agents. This represents a challenge for approaches such as imitation learning, which generally require expert demonstrations. We note prior work (e.g., (Fujimoto et al., 2018a; Kumar et al., 2019; Wu et al., 2019)) predominantly uses data with this property.

**Non-representable behavior policies, non-Markovian behavior policies, and partial observability.** When the dataset is generated from a partially-trained agent, we ensure that the behavior policy can be realized within our model class. However, real-life behavior may not originate from a policy within our model class, which can introduce additional representational errors. For example, data generated from human demonstrations or hand-crafted controllers may fall outside of the model class. More generally, **non-Markovian** policies and tasks with **partial observability** can introduce additional modeling errors when we estimate action probabilities under the assumption that the data was generated from a Markovian policy. These errors can cause additional bias for offline RL algorithms, especially in methods that assume access to action probabilities from a Markovian policy such as importance weighting (Precup et al., 2000).

**Realistic domains**. As we discussed previously, real-world evaluation is the ideal setting for benchmarking offline RL, however, it is at odds with a widely-accessible and reproducible benchmark. To strike a balance, we opted for simulated environments which have been previously studied and are broadly accepted by the research community. These simulation packages (such as MuJoCo, Flow, and CARLA) have been widely used to benchmark *online* RL methods and are known to fit well into that role. Moreover, on several domains we utilize human demonstrations or mathematical models of human behavior in order to provide datasets generated from realistic processes. However, this did have the effect of restricting our choice of tasks. While recent progress has been made on simulators for recommender systems (e.g., (Ie et al., 2019)), they use "stylized" user models and they have not been thoroughly evaluated by the community yet. In the future as simulators mature, we hope to include additional tasks.

In addition, we include a variety of qualitatively different tasks to provide broad coverage of the types of domains where offline RL could be used. We include locomotion, traffic management, autonomous driving, and robotics tasks. We also provide tasks with a wide range in difficulty, from tasks current algorithms can already solve to harder problems that are currently out of reach. Finally, for consistency with prior works, we also include the OpenAI Gym robotic locomotion tasks and similar datasets used by Fujimoto et al. (2018a); Kumar et al. (2019); Wu et al. (2019).

## 5 TASKS AND DATASETS

Given the properties outlined in Section 4, we assembled the following tasks and datasets. All tasks consist of an offline *dataset* (typically $10^6$ steps) of trajectory samples for training, and a *simulator* for evaluation. The mapping is not one-to-one – several tasks use the same simulator with different datasets. Appendix C lists domains and dataset types along with their sources and Appendix A contains a more comprehensive table of statistics such as size. Our code and datasets have been released open-source and are on our website at `https://sites.google.com/view/d4rl-anonymous/`.

**Maze2D.** *(Non-markovian policies, undirected and multitask data)*
The Maze2D domain is a navigation task requiring a 2D agent to reach a fixed goal location. The tasks are designed to provide a simple test of the ability of offline RL algorithms to stitch together previously collected subtrajectories to find the shortest path to the evaluation goal. Three

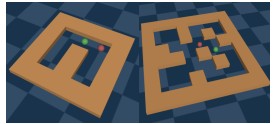

maze layouts are provided. The "umaze" and "medium" mazes are shown to the right, and the "large" maze is shown below.

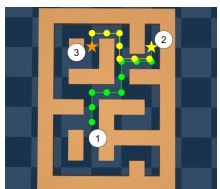

The data is generated by selecting goal locations at random and then using a planner that generates sequences of waypoints that are followed using a PD controller. In the figure on the left, the waypoints, represented by circles, are planned from the starting location (1) along the path to a goal (2). Upon reaching a threshold distance to a waypoint, the controller updates its internal state to track the next waypoint along the path to the goal. Once a goal is reached, a new goal is selected (3) and the process continues. The trajectories in the dataset are visualized in Appendix G. Because the controllers memorize the reached waypoints, the data collection policy is non-Markovian.

**AntMaze.** *(Non-markovian policies, sparse rewards, undirected and multitask data)* The AntMaze domain is a navigation domain that replaces the 2D ball from Maze2D with the more complex 8-DoF "Ant" quadraped robot. We introduce this domain to test the stitching challenge using a morphologically complex robot that could mimic real-world robotic navigation tasks. Additionally, for this task we use a sparse 0-1 reward which is activated upon reaching the goal.

The data is generated by training a goal reaching policy and using it in conjunction with the same high-level waypoint generator from **Maze2D** to provide subgoals that guide the agent to the goal. The same 3 maze layouts are used: "umaze", "medium", and "large". We introduce three flavors of datasets: 1) the ant is commanded to reach a specific goal from a fixed start location (`antmaze-umaze-v0`), 2) in the "diverse" datasets, the ant is commanded to a random goal from a random start location, 3) in

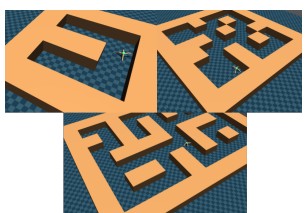

the "play" datasets, the ant is commanded to specific hand-picked locations in the maze (which are not necessarily the goal at evaluation), starting from a different set of hand-picked start locations. As in Maze2D, the controllers for this task are non-Markovian as they rely on tracking visited waypoints. Trajectories in the dataset are visualized in Appendix G.

**Gym-MuJoCo.** *(Suboptimal agents, narrow data distributions)* The Gym-MuJoCo tasks (Hopper, HalfCheetah, Walker2d) are popular benchmarks used in prior work in offline deep RL (Fujimoto et al., 2018a; Kumar et al., 2019; Wu et al., 2019). For consistency, we provide standardized datasets similar to previous work, and additionally propose mixing datasets to test the impact of heterogenous policy mixtures. We expect that methods that rely on regularizing to the behavior policy may fail when the data contains poorly performing trajectories.

The "medium" dataset is generated by first training a policy online using Soft Actor-Critic (Haarnoja et al., 2018a), early-stopping the training, and collecting 1M samples from this partially-trained policy. The "random" datasets are generated by unrolling a randomly initialized policy on these three domains. The

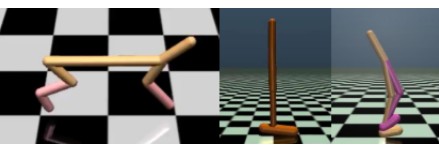

"medium-replay" dataset consists of recording all samples in the replay buffer observed during training until the policy reaches the "medium" level of performance. Datasets similar to these three have been used in prior work, but in order to evaluate algorithms on mixtures of policies, we further introduce a "medium-expert" dataset by mixing equal amounts of expert demonstrations and suboptimal data, generated via a partially trained policy or by unrolling a uniform-at-random policy.

**Adroit.** *(Non-representable policies, narrow data distributions, sparse rewards, realistic)* The Adroit domain (Rajeswaran et al., 2018) (pictured left) involves controlling a 24-DoF simulated Shadow Hand robot tasked with hammering a nail, opening a door, twirling a pen, or picking up and

moving a ball. This domain was selected to measure the effect of a narrow expert data distributions and human demonstrations on a sparse reward, high-dimensional robotic manipulation task.

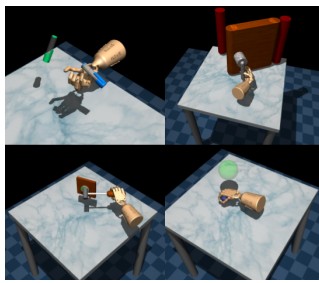

While Rajeswaran et al. (2018) propose utilizing human demonstrations, in conjunction with online RL fine-tuning, our benchmark adapts these tasks for evaluating the fully offline RL setting. We include three types of datasets for each task, two of which are included from the original paper: a small amount of demonstration data from a human ("human") (25 trajectories per task) and a large amount of expert data from a fine-tuned RL policy ("expert"). To mimic the use-case where a practitioner collects a small amount of additional data from a policy trained on the demonstrations, we introduce a third dataset generated by training an imitation policy on the demonstrations, running the policy, and mixing data at a 50-50 ratio with the demonstrations, referred to as "cloned." The Adroit domain has several unique properties that make it qualitatively different from the Gym MuJoCo tasks. First, the data is collected in from human demonstrators. Second, each task is difficult to solve with online RL, due to sparse rewards and exploration challenges, which make cloning and online RL alone insufficient. Lastly, the tasks are high dimensional, presenting a representation learning challenge.

**FrankaKitchen.** *(Undirected and multitask data, realistic)* The Franka Kitchen domain, first proposed by Gupta et al. (2019), involves controlling a 9-DoF Franka robot in a kitchen environment containing several common household items: a microwave, a kettle, an overhead light, cabinets, and an oven. The goal of each task is to interact with the items in order to reach a desired goal configuration. For example, one such state is to have the microwave and sliding cabinet door open with the kettle on the top burner and the overhead light on. This domain benchmarks the effect of multitask

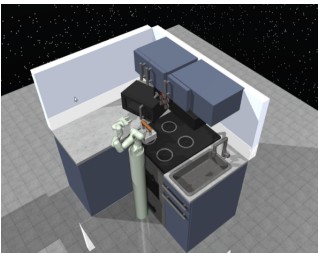

behavior on a realistic, non-navigation environment in which the "stitching" challenge is non-trivial because the collected trajectories are complex paths through the state space. As a result, algorithms must rely on generalization to unseen states in order to solve the task, rather than relying purely on trajectories seen during training.

In order to study the effect of "stitching" and generalization, we use 3 datasets of human demonstrations, originally proposed by Gupta et al. (2019). The "complete" dataset consists of the robot performing all of the desired tasks in order. This provides data that is easy for an imitation learning method to solve. The "partial" and "mixed" datasets consist of undirected data, where the robot performs subtasks that are not necessarily related to the goal configuration. In the "partial" dataset, a subset of the dataset is guaranteed to solve the task, meaning an imitation learning agent may learn by selectively choosing the right subsets of the data. The "mixed" dataset contains no trajectories which solve the task completely, and the RL agent must learn to assemble the relevant sub-trajectories. This dataset requires the highest degree of generalization in order to succeed.

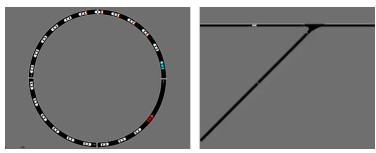

**Flow.** *(Non-representable policies, realistic)* The Flow benchmark (Vinitsky et al., 2018) is a framework for studying traffic control using deep reinforcement learning. We use two tasks in the Flow benchmark which involve controlling autonomous vehicles to maximize the flow of traffic through a ring or merge road configuration (left).

We use the Flow domain in order to provide a task that simulates real-world traffic dynamics. A large challenge in autonomous driving is to be able to directly learn from human behavior. Thus, we include "human" data from agents controlled by the intelligent driver model (IDM) (Treiber et al., 2000), a hand-designed model of human driving behavior. In order to provide data with a wider distribution as a reference, we also include "random" data generated from an agent that commands random vehicle accelerations.

**Offline CARLA.** *(Partial observability, non-representable policies, undirected and multitask data, realistic)* CARLA (Dosovitskiy et al., 2017) is a high-fidelity autonomous driving simulator

that has previously been used with imitation learning approaches (Rhinehart et al., 2018; Codevilla et al., 2018) from large, static datasets. The agent controls the throttle (gas pedal), the steering, and the break pedal for the car, and receives 48x48 RGB images from the driver's perspective as observations. We propose two tasks for offline RL: lane following within a figure eight path (shown to the right, top picture), and navigation within a small town (bottom picture). The principle challenge of the CARLA domain is partial observability and visual complexity, as all observations are provided as first-person RGB images.

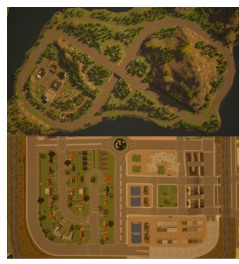

The datasets in both tasks are generated via hand-designed controllers meant to emulate human driving - the lane-following task uses simple heuristics to avoid cars and keep the car within lane boundaries, whereas the navigation task layers an additional high-level controller on top that takes turns randomly at intersections. Similarly to the Maze2D and AntMaze domains, this dataset consists of undirected navigation data in order to test the "stitching" property, however, it is in a more perceptually challenging domain.

**Evaluation protocol.** Previous work tunes hyperparameters with online evaluation inside the simulator, and as Wu et al. (2019) show, the hyperparameters have a large impact on performance. Unfortunately, extensive online evaluation is not practical in real-world applications and this leads to over optimistic performance expectations when the system is deployed in a truly offline setting. To rectify this problem, we designate a subset of tasks in each domain as "training" tasks, where hyperparameter tuning is allowed, and another subset as "evaluation" tasks on which final performance is measured (See Appendix D Table 5).

To facilitate comparison across tasks, we normalize scores for each environment roughly to the range between 0 and 100, by computing `normalized score` $= 100 * \frac{\texttt{score}-\texttt{random score}}{\texttt{expert score}-\texttt{random score}}$. A normalized score of 0 corresponds to the average returns (over 100 episodes) of an agent taking actions uniformly at random across the action space. A score of 100 corresponds to the average returns of a domain-specific expert. For Maze2D, and Flow domains, this corresponds to the performance of the hand-designed controller used to collect data. For CARLA, AntMaze, and FrankaKitchen, we used an estimate of the maximum score possible. For Adroit, this corresponds to a policy trained with behavioral cloning on human-demonstrations and fine-tuned with RL. For Gym-MuJoCo, this corresponds to a soft-actor critic (Haarnoja et al., 2018b) agent.

## 6    BENCHMARKING PRIOR METHODS

We evaluated recently proposed offline RL algorithms and several baselines on our offline RL benchmarks. This evaluation **(1)** provides baselines as a reference for future work, and **(2)** identifies shortcomings in existing offline RL algorithms in order to guide future research. The average normalized performance for all tasks is plotted in the figure to the right. It is clear that as we move beyond simple tasks and data collection strategies, differences between algorithms are exacerbated and deficiencies in all algorithms are revealed. See Appendix Table 2 for normalized results for all tasks, Appendix Table 3 for unnormalized scores, and Appendix E for experimental details.

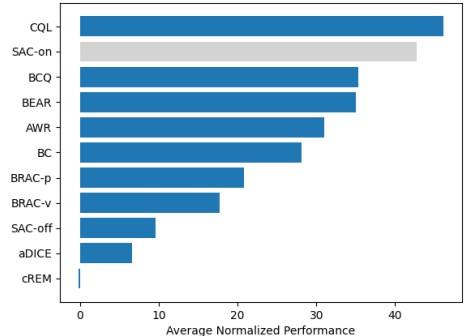

We evaluated behavioral cloning (BC), online and offline soft actor-critic (SAC) (Haarnoja et al., 2018b), bootstrapping error reduction (BEAR) (Kumar et al., 2019), and behavior-regularized actor-critic (BRAC) (Wu et al., 2019), advantage-weighted regression (AWR) (Peng et al., 2019), batch-constrained Q-learning (BCQ) (Fujimoto et al., 2018a), continuous action random ensemble mixtures (cREM) (Agarwal et al., 2019b), and AlgaeDICE (Nachum et al., 2019). We note that REM was originally designed for discrete action spaces, and the continuous action version has not been developed extensively. In most domains, we expect online SAC to outperform offline algorithms

when given the same amount of data because this baseline is able to collect on-policy data. There are a few exceptions, such as for environments where exploration challenges make it difficult to find high-reward states, such as the Adroit and maze domains.

Overall, the benchmarked algorithms obtained the most success on datasets generated from an RL-trained policy, such as in the Adroit and Gym-MuJoCo domains. In these domains, offline RL algorithms are able to match the behavior policy when given expert data, and outperform when given suboptimal data. This positive result is expected, as it is the predominant setting in which these prior algorithms have been benchmarked in past work.

Another positive result comes in the form of sparse reward tasks. In particular, many methods were able to outperform the baseline online SAC method on the Adroit and AntMaze domains. This indicates that offline RL is a promising paradigm for overcoming exploration challenges, which has also been confirmed in recent work (Nair et al., 2020). We also find that conservative methods that constrain the policy to the dataset, such as BEAR, AWR, CQL, and BCQ, are able to handle biased and narrow data distributions well on domains such as Flow and Gym-MuJoCo.

Tasks with undirected data, such as the Maze2D, FrankaKitchen, CARLA and AntMaze domains, are challenging for existing methods. Even in the simpler Maze2D domain, the large maze provides a surprising challenge for most methods. However, the smaller instances of Maze2D and AntMaze are very much within reach of current algorithms. Mixture distributions (a form of non-representable policies) were also challenging for all algorithms we evaluated. For MuJoCo, even though the medium-expert data contains expert data, the algorithms performed roughly on-par with medium datasets, except for hopper. The same pattern was found in the cloned datasets for Adroit, where the algorithms mostly performed on-par with the limited demonstration dataset, even though they had access to additional data.

We find that many algorithms were able to succeed to some extent on tasks with controller-generated data, such as Flow and carla-lane. We also note that tasks with limited data, such as human demonstrations in Adroit and FrankaKitchen, remain challenging. This potentially points to the need for more sample-efficient methods, as big datasets may not be always be available.

## 7 DISCUSSION

We have proposed an open-source benchmark for offline RL. The choice of tasks and datasets were motivated by properties reflected in real-world applications, such as narrow data distributions and undirected, logged behavior. Existing benchmarks have largely concentrated on robotic control using data produced by policies trained with RL (Fujimoto et al., 2018a; Kumar et al., 2019; Wu et al., 2019; Gulcehre et al., 2020; Dulac-Arnold et al., 2020). This can give a misleading sense of progress, as we have shown in our experiments that many of the more challenging properties that we expect real-world datasets to have appear to result in a substantial challenge for existing methods.

We believe that offline RL holds great promise as a potential paradigm to leverage vast amounts of existing sequential data within the flexible decision making framework of reinforcement learning. We hope that providing a benchmark that is representative of potential problems in offline RL, but that still can be accessibly evaluated in simulation, will greatly accelerate progress in this field and create new opportunities to apply RL in many real-world application areas.

There are several important properties exhibited in some real-world applications of RL that are not explored in-depth in our benchmark. One is stochasticity of the environment, which can occur in systems such as financial markets, healthcare, or advertisement. And while we explore domains with large observation spaces, some domains such recommender systems can exhibit large action spaces. Finally, our tasks are predominantly in the robotics, autonomous driving, and traffic control. There are several other areas (e.g. in finance, industrial, or operations research) which could provide simulated, yet still challenging, benchmarks.

Ultimately, we would like to see offline RL applications move from simulated domains to real-world domains, using real-world datasets. This includes exciting areas such as recommender systems, where user behavior can be easily logged, and medicine, where doctors must keep complete medical records for each patient. A key challenge in these domains is that reliable evaluation must be done in a *real system* or using off-policy evaluation (OPE) methods. We believe that both reliable OPE methods and real-world benchmarks which are standardized across different research groups, will be important to establish for future benchmarks in offline RL.

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

# Appendices

## A  TASK PROPERTIES

The following is a full list of task properties and dataset statistics for all tasks in the benchmark. Note that the full dataset for "carla-town" requires over 30GB of memory to store, so we also provide a subsampled version of the dataset which we used in our experiments.

| Domain | Task Name | Controller Type | # Samples |
|---|---|---|---|
| Maze2D | maze2d-umaze | Planner | $10^6$ |
| | maze2d-medium | Planner | $2 * 10^6$ |
| | maze2d-large | Planner | $4 * 10^6$ |
| AntMaze | antmaze-umaze | Planner | $10^6$ |
| | antmaze-umaze-diverse | Planner | $10^6$ |
| | antmaze-medium-play | Planner | $10^6$ |
| | antmaze-medium-diverse | Planner | $10^6$ |
| | antmaze-large-play | Planner | $10^6$ |
| | antmaze-large-diverse | Planner | $10^6$ |
| Gym-MuJoCo | hopper-random | Policy | $10^6$ |
| | hopper-medium | Policy | $10^6$ |
| | hopper-medium-replay | Policy | 200920 |
| | hopper-medium-expert | Policy | $2 \times 10^6$ |
| | halfcheetah-random | Policy | $10^6$ |
| | halfcheetah-medium | Policy | $10^6$ |
| | halfcheetah-medium-replay | Policy | 101000 |
| | halfcheetah-medium-expert | Policy | $2 \times 10^6$ |
| | walker2d-random | Policy | $10^6$ |
| | walker2d-medium | Policy | $10^6$ |
| | walker2d-medium-replay | Policy | 100930 |
| | walker2d-medium-expert | Policy | $2 \times 10^6$ |
| Adroit | pen-human | Human | 5000 |
| | pen-cloned | Policy | $5 * 10^5$ |
| | pen-expert | Policy | $5 * 10^5$ |
| | hammer-human | Human | 11310 |
| | hammer-cloned | Policy | $10^6$ |
| | hammer-expert | Policy | $10^6$ |
| | door-human | Human | 6729 |
| | door-cloned | Policy | $10^6$ |
| | door-expert | Policy | $10^6$ |
| | relocate-human | Human | 9942 |
| | relocate-cloned | Policy | $10^6$ |
| | relocate-expert | Policy | $10^6$ |
| Flow | flow-ring-random | Policy | $10^6$ |
| | flow-ring-controller | Policy | $10^6$ |
| | flow-merge-random | Policy | $10^6$ |
| | flow-merge-controller | Policy | $10^6$ |
| FrankaKitchen | kitchen-complete | Policy | 3680 |
| | kitchen-partial | Policy | 136950 |
| | kitchen-mixed | Policy | 136950 |
| CARLA | carla-lane | Planner | $10^5$ |
| | carla-town | Planner | $2 * 10^6$ full |
| | | | $10^5$ subsampled |

Table 1: Statistics for each task in the benchmark. For the controller type, "planner" refers to a hand-designed navigation planner, "human" refers to human demonstrations, and "policy" refers to random or neural network policies. The number of samples refers to the number of environment transitions recorded in the dataset.

# B RESULTS BY DOMAIN

The following tables summarize performance for each domain (excluding CARLA, due to all algorithms performing poorly), sorted by the best performing algorithm to the worst.

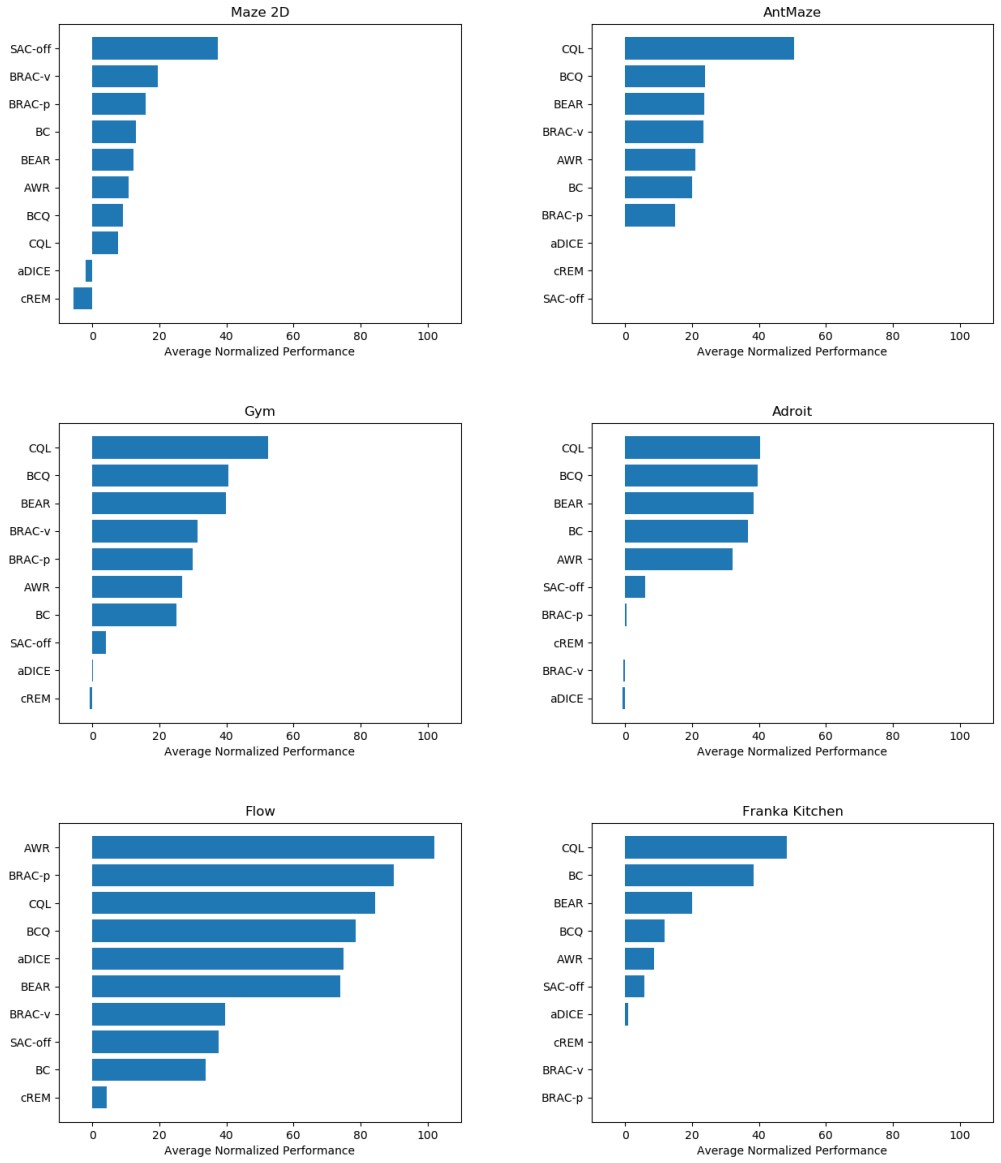

| | Task Name | SAC | BC | SAC-off | BEAR | BRAC-p | BRAC-v | AWR | cREM | BCQ | aDICE | CQL |
|---|---|---|---|---|---|---|---|---|---|---|---|---|
| Maze 2D | maze2d-umaze | 62.7 | 3.8 | 88.2 | 3.4 | 4.7 | -16.0 | 1.0 | -15.8 | 12.8 | -15.7 | 5.7 |
| | maze2d-medium | 21.3 | 30.3 | 26.1 | 29.0 | 32.4 | 33.8 | 7.6 | 0.9 | 8.3 | 10.0 | 5.0 |
| | maze2d-large | 2.7 | 5.0 | -1.9 | 4.6 | 10.4 | 40.6 | 23.7 | -2.2 | 6.2 | -0.1 | 12.5 |
| AntMaze | antmaze-umaze | 0.0 | 65.0 | 0.0 | 73.0 | 50.0 | 70.0 | 56.0 | 0.0 | 78.9 | 0.0 | 74.0 |
| | antmaze-umaze-diverse | 0.0 | 55.0 | 0.0 | 61.0 | 40.0 | 70.0 | 70.3 | 0.0 | 55.0 | 0.0 | 84.0 |
| | antmaze-medium-play | 0.0 | 0.0 | 0.0 | 0.0 | 0.0 | 0.0 | 0.0 | 0.0 | 0.0 | 0.0 | 61.2 |
| | antmaze-medium-diverse | 0.0 | 0.0 | 0.0 | 8.0 | 0.0 | 0.0 | 0.0 | 0.0 | 0.0 | 0.0 | 53.7 |
| | antmaze-large-play | 0.0 | 0.0 | 0.0 | 0.0 | 0.0 | 0.0 | 0.0 | 0.0 | 6.7 | 0.0 | 15.8 |
| | antmaze-large-diverse | 0.0 | 0.0 | 0.0 | 0.0 | 0.0 | 0.0 | 0.0 | 0.0 | 2.2 | 0.0 | 14.9 |
| Gym | halfcheetah-random | 100.0 | 2.1 | 30.5 | 25.1 | 24.1 | 31.2 | 2.5 | -2.6 | 2.2 | -0.3 | 35.4 |
| | walker2d-random | 100.0 | 1.6 | 4.1 | 7.3 | -0.2 | 1.9 | 1.5 | -0.3 | 4.9 | 0.5 | 7.0 |
| | hopper-random | 100.0 | 9.8 | 11.3 | 11.4 | 11.0 | 12.2 | 10.2 | 0.7 | 10.6 | 0.9 | 10.8 |
| | halfcheetah-medium | 100.0 | 36.1 | -4.3 | 41.7 | 43.8 | 46.3 | 37.4 | -2.6 | 40.7 | -2.2 | 44.4 |
| | walker2d-medium | 100.0 | 6.6 | 0.9 | 59.1 | 77.5 | 81.1 | 17.4 | -0.2 | 53.1 | 0.3 | 79.2 |
| | hopper-medium | 100.0 | 29.0 | 0.8 | 52.1 | 32.7 | 31.1 | 35.9 | 0.6 | 54.5 | 1.2 | 58.0 |
| | halfcheetah-medium-replay | 100.0 | 38.4 | -2.4 | 38.6 | 45.4 | 47.7 | 40.3 | -3.0 | 38.2 | -2.1 | 46.2 |
| | walker2d-medium-replay | 100.0 | 11.3 | 1.9 | 19.2 | -0.3 | 0.9 | 15.5 | -0.2 | 15.0 | 0.6 | 26.7 |
| | hopper-medium-replay | 100.0 | 11.8 | 3.5 | 33.7 | 0.6 | 0.6 | 28.4 | 0.8 | 33.1 | 1.1 | 48.6 |
| | halfcheetah-medium-expert | 100.0 | 35.8 | 1.8 | 53.4 | 44.2 | 41.9 | 52.7 | -2.6 | 64.7 | -0.8 | 62.4 |
| | walker2d-medium-expert | 100.0 | 6.4 | -0.1 | 40.1 | 76.9 | 81.6 | 53.8 | -0.2 | 57.5 | 0.4 | 111.0 |
| | hopper-medium-expert | 100.0 | 111.9 | 1.6 | 96.3 | 1.9 | 0.8 | 27.1 | 0.7 | 110.9 | 1.1 | 98.7 |
| Adroit | pen-human | 21.6 | 34.4 | 6.3 | -1.0 | 8.1 | 0.6 | 12.3 | 3.5 | 68.9 | -3.3 | 37.5 |
| | hammer-human | 0.2 | 1.5 | 0.5 | 0.3 | 0.3 | 0.2 | 1.2 | 0.2 | 0.5 | 0.3 | 4.4 |
| | door-human | -0.2 | 0.5 | 3.9 | -0.3 | -0.3 | -0.3 | 0.4 | -0.1 | -0.0 | -0.0 | 9.9 |
| | relocate-human | -0.2 | 0.0 | 0.0 | -0.3 | -0.3 | -0.3 | -0.0 | -0.2 | -0.1 | -0.1 | 0.2 |
| | pen-cloned | 21.6 | 56.9 | 23.5 | 26.5 | 1.6 | -2.5 | 28.0 | -3.4 | 44.0 | -2.9 | 39.2 |
| | hammer-cloned | 0.2 | 0.8 | 0.2 | 0.3 | 0.3 | 0.3 | 0.4 | 0.2 | 0.4 | 0.3 | 2.1 |
| | door-cloned | -0.2 | -0.1 | 0.0 | -0.1 | -0.1 | -0.1 | 0.0 | -0.1 | 0.0 | 0.0 | 0.4 |
| | relocate-cloned | -0.2 | -0.1 | -0.2 | -0.3 | -0.3 | -0.3 | -0.2 | -0.2 | -0.3 | -0.3 | -0.1 |
| | pen-expert | 21.6 | 85.1 | 6.1 | 105.9 | -3.5 | -3.0 | 111.0 | 0.3 | 114.9 | -3.5 | 107.0 |
| | hammer-expert | 0.2 | 125.6 | 25.2 | 127.3 | 0.3 | 0.3 | 39.0 | 0.2 | 107.2 | 0.3 | 86.7 |
| | door-expert | -0.2 | 34.9 | 7.5 | 103.4 | 0.3 | -0.3 | 102.9 | -0.2 | 99.0 | 0.0 | 101.5 |
| | relocate-expert | -0.2 | 101.3 | -0.3 | 98.6 | -0.3 | -0.4 | 91.5 | -0.1 | 41.6 | -0.1 | 95.0 |
| Flow | flow-ring-controller | 100.7 | -57.0 | 9.2 | 62.7 | -12.3 | -91.2 | 75.2 | -47.4 | 76.2 | 15.2 | 52.0 |
| | flow-ring-random | 100.7 | 94.9 | 70.0 | 103.5 | 95.7 | 78.6 | 80.4 | -87.4 | 94.6 | 83.6 | 87.9 |
| | flow-merge-controller | 121.5 | 114.1 | 111.6 | 150.4 | 129.8 | 143.9 | 152.7 | 183.2 | 114.8 | 196.4 | 157.2 |
| | flow-merge-random | 121.5 | -17.1 | -40.1 | -20.6 | 146.2 | 27.3 | 99.6 | -31.2 | 28.2 | 4.7 | 40.6 |
| Franka Kitchen | kitchen-complete | 0.0 | 33.8 | 15.0 | 0.0 | 0.0 | 0.0 | 0.0 | 0.0 | 8.1 | 0.0 | 43.8 |
| | kitchen-partial | 0.6 | 33.8 | 0.0 | 13.1 | 0.0 | 0.0 | 15.4 | 0.0 | 18.9 | 0.0 | 49.8 |
| | kitchen-mixed | 0.0 | 47.5 | 2.5 | 47.2 | 0.0 | 0.0 | 10.6 | 0.0 | 8.1 | 2.5 | 51.0 |
| CARLA | carla-lane | -0.8 | 31.8 | 0.1 | -0.2 | 18.2 | 19.6 | -0.4 | 0.1 | -0.1 | -1.2 | 20.9 |
| | carla-town | 1.4 | -1.8 | -1.8 | -2.7 | -4.6 | -2.6 | 1.9 | -0.9 | 1.9 | -11.2 | -2.6 |

Table 2: Normalized results comparing online & offline SAC (SAC, SAC-off), bootstrapping error reduction (BEAR), behavior-regularized actor critic with policy (BRAC-p) or value (BRAC-v) regularization, behavioral cloning (BC), advantage-weighted regression (AWR), batch-constrained Q-learning (BCQ), continuous random ensemble mixtures (cREM), and AlgaeDICE (aDICE). Average results are reported over 3 seeds, and normalized to a score between 0 (random) and 100 (expert).

| Domain | Task Name | SAC | BC | SAC-off | BEAR | BRAC-p | BRAC-v | AWR | cREM | BCQ | AlgaeDICE | CQL |
|---|---|---|---|---|---|---|---|---|---|---|---|---|
| Maze2D | maze2d-umaze | 110.4 | 29.0 | 145.6 | 28.6 | 30.4 | 1.7 | 25.2 | 2.1 | 41.5 | 2.2 | 31.7 |
| | maze2d-medium | 69.5 | 93.2 | 82.0 | 89.8 | 98.8 | 102.4 | 33.2 | 15.6 | 35.0 | 39.6 | 26.4 |
| | maze2d-large | 14.1 | 20.1 | 1.5 | 19.0 | 34.5 | 115.2 | 70.1 | 0.8 | 23.2 | 6.5 | 40.0 |
| AntMaze | antmaze-umaze | 0.0 | 0.7 | 0.0 | 0.7 | 0.5 | 0.7 | 0.6 | 0.0 | 0.8 | 0.0 | 0.7 |
| | antmaze-umaze-diverse | 0.0 | 0.6 | 0.0 | 0.6 | 0.4 | 0.7 | 0.7 | 0.0 | 0.6 | 0.0 | 0.8 |
| | antmaze-medium-play | 0.0 | 0.0 | 0.0 | 0.0 | 0.0 | 0.0 | 0.0 | 0.0 | 0.0 | 0.0 | 0.6 |
| | antmaze-medium-diverse | 0.0 | 0.0 | 0.0 | 0.1 | 0.0 | 0.0 | 0.0 | 0.0 | 0.0 | 0.0 | 0.5 |
| | antmaze-large-play | 0.0 | 0.0 | 0.0 | 0.0 | 0.0 | 0.0 | 0.0 | 0.0 | 0.1 | 0.0 | 0.2 |
| | antmaze-large-diverse | 0.0 | 0.0 | 0.0 | 0.0 | 0.0 | 0.0 | 0.0 | 0.0 | 0.0 | 0.0 | 0.1 |
| Gym | halfcheetah-random | 12135.0 | -17.9 | 3502.0 | 2831.4 | 2713.6 | 3590.1 | 36.3 | -597.6 | -1.3 | -318.0 | 4114.8 |
| | walker2d-random | 4592.3 | 73.0 | 192.0 | 336.3 | -7.2 | 87.4 | 71.5 | -12.7 | 228.0 | 26.5 | 322.9 |
| | hopper-random | 3234.3 | 299.4 | 347.7 | 349.9 | 337.5 | 376.3 | 312.4 | 4.0 | 323.9 | 10.0 | 331.2 |
| | halfcheetah-medium | 12135.0 | 4196.4 | -808.6 | 4897.0 | 5158.8 | 5473.8 | 4366.1 | -600.0 | 4767.9 | -551.6 | 5232.1 |
| | walker2d-medium | 4592.3 | 304.8 | 44.2 | 2717.0 | 3559.9 | 3725.8 | 800.7 | -9.3 | 2441.0 | 15.5 | 2664.2 |
| | hopper-medium | 3234.3 | 923.5 | 5.7 | 1674.5 | 1044.0 | 990.4 | 1149.5 | 0.3 | 1752.4 | 17.5 | 2557.3 |
| | halfcheetah-medium-replay | 12135.0 | 4492.1 | -581.3 | 4517.9 | 5350.8 | 5640.6 | 4727.4 | -655.7 | 4463.9 | -540.8 | 5455.6 |
| | walker2d-medium-replay | 4592.3 | 518.6 | 87.8 | 883.8 | -11.5 | 44.5 | 712.5 | -8.7 | 688.7 | 31.0 | 1227.3 |
| | hopper-medium-replay | 3234.3 | 364.4 | 93.3 | 1076.8 | 0.5 | -0.8 | 904.0 | 5.8 | 1057.8 | 14.0 | 1227.3 |
| | halfcheetah-medium-expert | 12135.0 | 4169.4 | -55.7 | 6349.6 | 5208.1 | 4926.6 | 6267.3 | -600.8 | 7750.8 | -377.6 | 7466.9 |
| | walker2d-medium-expert | 4592.3 | 297.0 | -5.1 | 1842.7 | 3533.1 | 3747.5 | 2469.7 | -8.0 | 2640.3 | 19.7 | 5097.3 |
| | hopper-medium-expert | 3234.3 | 3621.2 | 32.9 | 3113.5 | 42.6 | 5.1 | 862.0 | 3.1 | 3588.5 | 15.5 | 3192.0 |
| Adroit | pen-human | 739.3 | 1121.9 | 284.8 | 66.3 | 339.0 | 114.7 | 463.1 | 199.7 | 2149.0 | -0.7 | 1214.0 |
| | hammer-human | -248.7 | -82.4 | -214.2 | -242.0 | -239.7 | -243.8 | -115.3 | -243.2 | -210.5 | -234.8 | 300.2 |
| | door-human | -61.8 | -41.7 | 57.2 | -66.4 | -66.5 | -66.4 | -44.4 | -58.7 | -56.6 | -56.5 | 234.3 |
| | relocate-human | -13.7 | -5.6 | -4.5 | -18.9 | -19.7 | -19.7 | -7.2 | -14.1 | -8.6 | -10.8 | 2.0 |
| | pen-cloned | 739.3 | 1791.8 | 797.6 | 885.4 | 143.4 | 22.2 | 931.3 | -6.5 | 1407.8 | 8.6 | 1264.6 |
| | hammer-cloned | -248.7 | -175.1 | -244.1 | -241.1 | -236.7 | -236.9 | -226.9 | -245.8 | -224.4 | -233.1 | -0.41 |
| | door-cloned | -61.8 | -60.7 | -56.3 | -60.9 | -58.7 | -59.0 | -56.1 | -58.0 | -56.3 | -56.4 | -44.76 |
| | relocate-cloned | -13.7 | -10.1 | -16.1 | -17.6 | -19.8 | -19.4 | -16.6 | -16.9 | -17.5 | -18.8 | -10.66 |
| | pen-expert | 739.3 | 2633.7 | 277.4 | 3254.1 | -7.8 | 6.4 | 3406.0 | 104.0 | 3521.3 | -6.9 | 3286.2 |
| | hammer-expert | -248.7 | 16140.8 | 3019.5 | 16359.7 | -241.4 | -241.1 | 4822.9 | -247.1 | 13731.5 | -235.2 | 11062.4 |
| | door-expert | -61.8 | 969.4 | 163.8 | 2980.1 | -66.4 | -66.6 | 2964.5 | -60.9 | 2850.7 | -56.5 | 2926.8 |
| | relocate-expert | -13.7 | 4289.3 | -18.2 | 4173.8 | -20.6 | -21.4 | 3875.5 | -11.7 | 1759.6 | -8.7 | 4019.9 |
| Flow | flow-ring-controller | 25.8 | -273.3 | -147.7 | -46.3 | -188.5 | -338.2 | -22.6 | -255.0 | -20.7 | -136.3 | -66.5 |
| | flow-ring-random | 25.8 | 14.7 | -32.4 | 31.0 | 16.2 | -16.2 | -12.7 | -330.9 | 14.2 | -6.8 | 15.1 |
| | flow-merge-controller | 375.4 | 359.8 | 354.6 | 436.5 | 392.9 | 422.9 | 441.4 | 505.8 | 361.4 | 533.9 | 450.9 |
| | flow-merge-random | 375.4 | 82.6 | 33.9 | 75.1 | 427.6 | 176.3 | 329.3 | 52.7 | 178.3 | 128.7 | 204.6 |
| FrankaKitchen | kitchen-complete | 0.0 | 1.4 | 0.6 | 0.0 | 0.0 | 0.0 | 0.0 | 0.0 | 0.3 | 0.0 | 1.8 |
| | kitchen-partial | 0.0 | 1.4 | 0.0 | 0.5 | 0.0 | 0.0 | 0.6 | 0.0 | 0.8 | 0.0 | 1.9 |
| | kitchen-mixed | 0.0 | 1.9 | 0.1 | 1.9 | 0.0 | 0.0 | 0.4 | 0.0 | 0.3 | 0.1 | 2.0 |
| Offline CARLA | carla-lane | -8.6 | 324.7 | -0.3 | -3.0 | 186.1 | 199.6 | -4.5 | -0.0 | -1.4 | -13.4 | 213.2 |
| | carla-town | -79.7 | -161.5 | -159.9 | -182.5 | -231.6 | -181.5 | -65.8 | -137.4 | -66.6 | -400.2 | -180.379 |

Table 3: The raw, un-normalized scores for each task and algorithm are reported in the table below. These scores represent the undiscounted return obtained from executing a policy in the simulator, averaged over 3 random seeds.

## C  TASK AND DATASETS

The following table lists the tasks and dataset types included in the benchmark, including sources for each.

| Domain | Source | Dataset Types |
|---|---|---|
| Maze2D | N/A | UMaze, Medium, Large |
| AntMaze | N/A | UMaze, Diverse, Play |
| Gym-MuJoCo | Brockman et al. (2016) Todorov et al. (2012) | Expert, Random, Medium (Various) Medium-Expert, Medium-Replay |
| Adroit | Rajeswaran et al. (2018) | Human, Expert (Rajeswaran et al., 2018) Cloned |
| Flow | Wu et al. (2017) | Random, Controller |
| Franka Kitchen | Gupta et al. (2019) | Complete, Partial, Mixed (Gupta et al., 2019) |
| CARLA | Dosovitskiy et al. (2017) | Controller |

Table 4:  Domains and dataset types contained within our benchmark.  Maze2D and AntMaze are new domains we propose.  For each dataset, we also include references to the source if originally proposed in another work.  Datasets borrowed from prior work include MuJoCo (Expert, Random, Medium), Adroit (Human, Expert), and FrankaKitchen (Complete, Partial, Mixed).  All other datasets are datasets proposed by this work.

## D  TRAINING AND EVALUATION TASK SPLIT

The following table lists our recommended protocol for hyperparameter tuning.  Hyperparameters should be tuned on the tasks listed on the left in the "Training" column, and algorithms should be evaluated without tuning on the tasks in the right column labeled "Evaluation".

| Domain | Training | Evaluation |
|---|---|---|
| Maze2D | maze2d-umaze maze2d-medium maze2d-large | maze2d-eval-umaze maze2d-eval-medium maze2d-eval-large |
| AntMaze | ant-umaze ant-umaze-diverse ant-medium-play ant-medium-diverse ant-large-play ant-large-diverse | ant-eval-umaze ant-eval-umaze-diverse ant-eval-medium-play ant-eval-medium-diverse ant-eval-large-play ant-eval-large-diverse |
| Adroit | pen-human pen-cloned pen-expert door-human door-cloned door-expert | hammer-human hammer-cloned hammer-expert relocate-human relocate-cloned relocate-expert |
| Gym | halfcheetah-random halfcheetah-medium halfcheetah-mixed halfcheetah-medium-expert walker2d-random walker2d-medium walker2d-mixed walker2d-medium-expert | hopper-random hopper-medium hopper-mixed hopper-medium-expert ant-random ant-medium ant-mixed and-medium-expert |

Table 5: Our recommended partition of tasks into "training" tasks where hyperparameter tuning is allowed, and "evaluation" tasks where final algorithm performance should be reported.

## E    EXPERIMENT DETAILS

For all experiments, we used default hyperparameter settings and minimal modifications to public implementations wherever possible, using 500K training iterations or gradient steps. The code bases we used for evaluation are listed below. The most significant deviation from original published algorithms was that we used an unofficial continuous-action implementation of REM (Agarwal et al., 2019b), which was originally implemented for discrete action spaces. We ran our experiments using Google cloud platform (GCP) on `n1-standard-4` machines.

- BRAC and AlgaeDICE: `https://github.com/google-research/google-research`
- AWR: `https://github.com/xbpeng/awr`
- SAC: `https://github.com/vitchyr/rlkit`
- BEAR: `https://github.com/aviralkumar2907/BEAR`
- Continuous-action REM: `https://github.com/theSparta/off_policy_mujoco`
- BCQ: `https://github.com/sfujim/BCQ`
- CQL: `https://github.com/aviralkumar2907/CQL`

## F    ASSESSING THE FEASIBILITY OF HARD TASKS

Few prior methods were able to successfully solve carla-town or the larger AntMaze tasks. While including tasks that present a challenge for current methods is important to ensure that our benchmark has room for improvement, it is also important to provide some confidence that the tasks are actually solvable. We verified this in two ways. First, we ensured that the trajectories observed in these tasks have adequate coverage of the state space. An illustration of the trajectories in the CARLA and AntMaze tasks are shown below, where trajectories are shown as different colored lines and the goal state is marked with a star. We see that in carla-town and AntMaze, each lane or corridor is traversed multiple times by the agent.

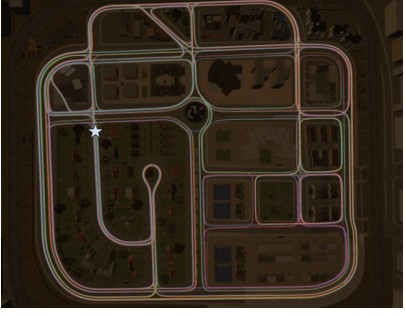

(a) carla-town

(b) antmaze-large    (c) maze2d-large

Second, the data in AntMaze was generated by having the ant follow the same high-level planner in the maze as in Maze2D. Because Maze2D is solvable by most methods, we would expect this to mean that AntMaze is potentially solvable as well. This While the dynamics of the ant itself are much more complex, its walking gait is a relatively regular periodic motion, and since the high-level waypoints are similar, we would expect the AntMaze data to provide similar coverage as in the 2D mazes, as shown in the figures on the right. While the Ant has more erratic motion, both datasets cover the the majority of the maze thoroughly. A comparison of the state coverage between Maze2D and AntMaze on all tasks is shown in the following Appendix section G.

## G   MAZE DOMAIN TRAJECTORIES

In this section, we visualized trajectories for the datasets in the Maze2D and AntMaze domains. Each image plots the states visited along each trajectory as a different colored line, overlaid on top of the maze. The goal state is marked as a white star.

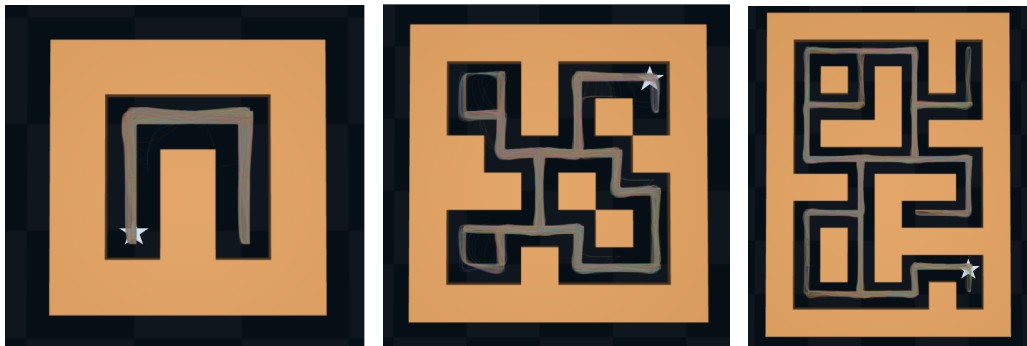

Figure 4: Trajectories visited in the Maze2D domain. From left-to-right: maze2d-umaze, maze2d-medium, and maze2d-large.

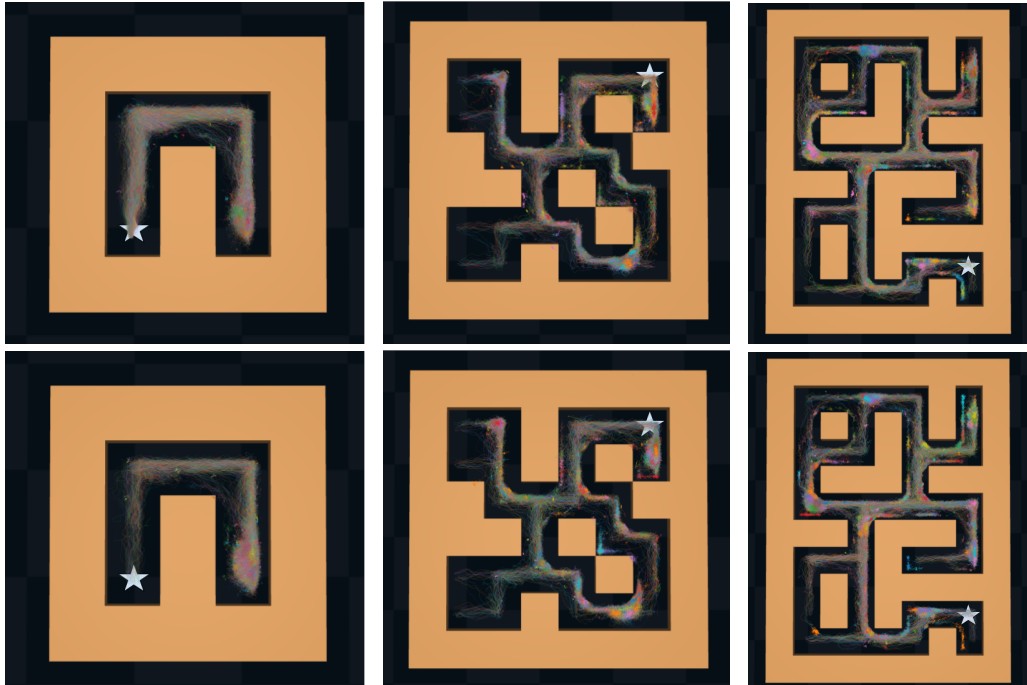

Figure 5: Trajectories visited in the AntMaze domain. Top row, from left-to-right: antmaze-umaze, antmaze-medium-play, and antmaze-large-play. Bottom row, from left-to-right: antmaze-umaze-diverse, antmaze-medium-diverse, and antmaze-large-diverse.

