# OpenReview forum: "D4RL: Datasets for Deep Data-Driven Reinforcement Learning"
_ICLR.cc/2021/Conference — Reject_

### Official Review · AnonReviewer1 · 2020-10-18
**A collection of offline datasets**

**Rating:** 2
**Confidence:** 5

**Review:**

This paper offers a new set of challenges for batch reinforcement learning coupled with a set of benchmarks, including autonomous robotics and driving domains. All domains also come with simulation environments.

While the dataset provided by the authors may be a useful resource for researchers in offline-RL, this paper does not introduce any new or novel ideas. Overall the main contribution of this work is the gathering of offline data in one place, reducing the time needed for other researchers to do so. It does not seem as if the authors did any non-trivial work other than annotating the data. Furthermore, there have been many previous work which have already collected offline datasets as part of their work.

I do not underestimate the importance the authors' hard work, nor the importance of the provided datasets to the RL community. Nevertheless, I do not believe this work should be published in a high-end conference without presenting any ideas that are not trivial or known to other researches.

The authors propose to use the following design factors in their offline datasets: narrow distributions, multi-task data, sparse rewards, suboptimal data, and partially observable policies. While these factors may indeed be good for testing offline-RL algorithms, they do not provide a complete picture of real world datasets. Real datasets present many more real-world problems, including:
1. Non-stationary data. Many real datasets are non-stationary. The non-stationary behavior could be mimicked or simulated from real behavior.
2. Real policies. Real datasets don't involve policies that were trained by RL agents. The authors could create datasets that are constructed by real human beings (for example, humans playing atari games, with mixed or different expertise). In a controlled setup, the datasets could be constructed so that policies are categorized (e.g., "level of non Markovianess`").
3. Causal structures. Real policies may act according to some causal structure in the background that is not necessarily known.
4. Reduction from real datasets. One could collect or use large amounts of high quality datasets from the real world and add certain corruptions to the data as to lower its quality (e.g., removing certain trajectories). If the initial data is of high quality, the corrupted data could be controlled well.
5. Changing and/or very large action sets. Real world datasets have changing and large datasets. As an example, consider ad placement, or text based tasks.
6. Non-robotic environments, including games but also real world problems.

If the authors bring together datasets that generate original ideas that have never been previously explored, then I believe it's more likely that this paper could be accepted in future venues.

---

> ### Public Comment · ~Yue_Wu17 · 2020-11-13
> **Some notes on the contribution**
>
> I am not related to the authors by any means, but I have used this dataset extensively. From my experience, this is one of the best datasets I have used, with much better quality and ease-of-access than a some that are published in well-known conferences and venues.
>
> The dataset also highlighted existing domain gaps and may generate new research questions: 1) sparse reward/temporal dependence regarding the maze environment. 2) sparse demonstrations regarding the human demonstrations in Adroit hand environment.  Both benchmarks show current works failing.
>
> I believe that the dataset will be greatly beneficial to the community. A top venue conference is not only a place for novel ideas, but also for researchers to exchange ideas and spread influence.

---

> > ### Comment · AnonReviewer1 · 2020-11-14
> > **Thank you for your comment**
> >
> > I respectfully disagree. While there is no doubt that this paper may benefit many researchers, its main contribution is the reduction of time it would take them to create such datasets.
> >
> > With regard to research questions: I didn't find anything in the paper that I had not already read in previous work.
> >
> > As an additional note I am also worried about D4RL becoming a standardized benchmark for offline reinforcement learning as its benchmarks are in fact far from real data behavior, especially in the behvioral (policy) aspect.
> >
> > Finally, regarding your remark on the purpose of top venue conferences, I will not argue as I believe this is open for interpretation and discussion. Nevertheless, I stand by my opinion.

---

> > > ### Comment · AnonReviewer4 · 2020-11-14
> > > **The dilemma leads to controversy**
> > >
> > > In this very controversial discussion I would like to express my support for both reviewer 1, who criticizes the benchmark suite's poor orientation towards real problems, and reviewer 2, who criticizes the use of non-free software for a proposed benchmark suite, by stating that I fully understand these arguments. As the title of my own review "Premature or high time?" indicates, I see the dilemma that on the one hand there is a demand for a benchmark suit for offline RL, on the other hand the solution presented here has significant weaknesses. I will not insist on my original rating of 6, but currently see more arguments for reducing this score than to keep it.

---

> > > > ### Comment · AnonReviewer4 · 2020-11-24
> > > > **Still in favor of acceptance**
> > > >
> > > > At the current state of the discussion I no longer think about lowering my score.

---

> ### Author Response · Authors · 2020-11-16
> **Clarifications on contribution**
>
> Thank you for your comments and feedback. We respond to individual comments below. Please let us know if this addresses your concerns.
>
> > “If the authors bring together datasets that generate original ideas that have never been previously explored, then I believe it's more likely that this paper could be accepted in future venues.”
>
> We believe that benchmark and evaluation papers should be measured on their potential impact and insights provided. Constructing a systematic evaluation of existing methods, providing insights into their shortcomings to guide future research (Section 6), and implementing an easy-to-use research platform (see our website and code: https://sites.google.com/view/d4rl-anonymous/) are all contributions that further advance this field.
>
> D4RL has already facilitated standardized evaluation and benchmarking, and moving beyond reliance on MuJoCo locomotion tasks and datasets obtained from replay buffers of online RL algorithms. This work has been used as a starting point for novel ideas as evidenced by the rapid adoption of D4RL, and the number of papers submitted to this conference that use it as their primary evaluation task such as:
> - “Offline Policy Optimization with Variance Regularization”
> - “Uncertainty Weighted Offline Reinforcement Learning”
> - “Risk-Averse Offline Reinforcement Learning”
> - “Addressing Distribution Shift in Online Reinforcement Learning with Offline Datasets”
> - “BRAC+: Going Deeper with Behavior Regularized Offline Reinforcement Learning”
> - “Fine-Tuning Offline Reinforcement Learning with Model-Based Policy Optimization”
> - “EMaQ: Expected-Max Q-Learning Operator for Simple Yet Effective Offline and Online RL”
> - “OPAL: Offline Primitive Discovery for Accelerating Offline Reinforcement Learning”
> - “Model-Based Offline Planning”
>
>  And in NeurIPS 2020 papers:
> - “Conservative Q-Learning for Offline Reinforcement Learning”. Kumar et. al.
> - “Model-Based Offline Policy Optimization”. Yu et al.
> - “Continual Learning of Control Primitives: Skill Discovery via Reset-Games”. Xu et. al.
>
> > “Real datasets present many more real-world problems, including: Non-stationary data… Real policies… Causal structures… Reduction from real datasets… Changing and/or very large action sets… Non-robotic environments”
>
> While we agree that these are important properties of real-world datasets, no single benchmark can cover every real world problem. Our benchmark tasks provide coverage of several important real-world properties that we suspected have a large impact on offline RL and have not previously been evaluated in offline RL. For example, the Adroit/Franka domains contain real policies obtained by teleoperation. We have added a discussion of these properties and the limitations of our benchmark to emphasize their importance and future opportunity in Section 7.
>
> > “While the dataset provided by the authors may be a useful resource... this paper does not introduce any new or novel ideas. Overall the main contribution of this work is the gathering of offline data in one place”
>
> We agree that our primary contribution is not novel methods or analyses, however, you understate our contributions. In the Gym/Franka/Adroit domains, we do use existing datasets, however, for the remaining domains, we either construct datasets for tasks that have not previously been studied in offline RL or in the case of Maze2D/AntMaze, construct novel domains to study specific properties. Appendix C details which datasets we borrowed as part of the construction of this benchmark.
>
> Prior to our work, the predominant method for evaluating offline RL algorithms was to generate data from online RL agents on Atari or Gym tasks. As discussed in [2] and shown in our empirical evaluation, this does not exercise important dimensions of the problem. Our data generation procedures bring attention to important and neglected properties in offline RL (e.g., narrow data distributions, multitask data, and non-representable policies) that have a significant impact on performance and have already guided followup research.
>
> For example, recent work [1, 2] has provided contradicting evidence on the importance of the data generating distribution in offline RL. D4RL contains multiple domains where this problem can be explored in depth, across a wide variety of evaluated algorithms. Our empirical evaluations in Table 1 reveal clear performance differences under different data generation settings.
>
> Finally, we provide a systematic evaluation of 11 RL methods, including state-of-the-art algorithms. Previously, no such extensive evaluation on a common dataset had been done in offline RL. The standardization and thorough evaluation of a benchmark with appealing properties (including several that you listed as important real-world properties) is a valuable contribution.
>
> [1] “An Optimistic Perspective on Offline Reinforcement Learning” Agrawal et. al. 2020
>
> [2] “Behavior Regularized Offline Reinforcement Learning” Wu et. al. 2019

---

> > ### Author Response · Authors · 2020-11-20
> > **Additional concerns?**
> >
> > Hello, we appreciate the constructive feedback on the paper. We are wondering if our response addresses your concerns, or if you think additional changes are needed. Thank you!

---

> > ### Comment · AnonReviewer1 · 2020-11-22
> > **Response to rebuttal**
> >
> > Thank you for your response. I would like to reiterate that I acknowledge the importance of offline datasets for offline RL, and do not underestimate the time it took to construct these datasets. My score and views follow as I see a negative impact this work (at its current state) may have on future research in the field.
> >
> > Consider the analogy of creating simulated environments for online reinforcement learning. How should papers introducing such new environments be evaluated? There are two types of new environments that I see fit:
> > 1. synthetic environments: These types of environments are not supposed to represent anything in the real world. Their sole purpose is to demonstrate specific characteristics of MDPs, and allow researchers to test their algorithms in highly controlled experiments.
> > 2. simulated / real environments: These include computer games (such as ATARI) and simulators based on mathematical models (such as MUJOCO). Computer games are interesting to solve as we have human benchmarks to compare against (achieving super-human performance is interesting), whereas simulators based on mathematical models are interesting because they are a good proxy of the real world.
> >
> > Now, let us consider offline RL. In offline RL one of the main challenges revolves around the properties of the policy / policies that generated the data. Here, we could also divide datasets into two types:
> > 1. synthetic behavior datasets: These datasets are constructed by synthetic / simulated environments, yet the behavior that generated the data is synthetic. These types of datasets have two purposes: to understand the limits of offline-RL with relation to the environment, but also, importantly, to demonstrate specific characteristics of the behavior that generated the data.
> > 2. real behavior datasets: These datasets should incorporate some form of real behavior. Such datasets could include real human behavior, accepted mathematical models of behavior in real offline data, or any type of behavior that can be convincing to represent real behavior in the given environment. When human behavior is involved, causal and non-stationarity factors are important aspects that must be controlled in the data creation process.
> >
> > The authors are currently focusing their paper on real behavioral datasets, yet their datasets consist of synthetic behaviors. It is unclear to me why these behaviors represent anything realistic.
> >
> > On the other hand, the authors also don't generate good synthetic behaviors. Their behaviors are not reproducible, and it is unclear if the presented domains are the right place to test for synthetic problems.
> >
> >
> > ### My main concern:
> >
> > As the authors claim, there are various papers that readily use the D4RL datasets in their work. I do not find this as a reason to accept the paper. On the contrary, I find it worrying that these benchmarks may produce bad evaluations for offline RL algorithms. Aside from the missing (fundamental) problems of real data, the authors mainly focused on continuous control tasks and synthetic policies.
> >
> > I strongly believe that accepting this paper (at its current state) will have a negative impact on research in offline RL. The authors should explicitly focus either on synthetic behavior or real behavior. In case of synthetic datasets, the authors should provide clean, simple datasets (even tabular), make the behaviors easy to reproduce, and make a notion of optimality clear. If the authors' focus is on realistic datasets they should span out of simple continuous control domains, characterize their generated policies better, use real humans to generate policies, and take into account causal + non stationarity factors.

---

> > > ### Author Response · Authors · 2020-11-22
> > > **On the use of real data**
> > >
> > > Thank you for the reply, and you raise good points. We appreciate your comment about the importance of realistic evaluation domains, and we agree that (1) synthetic datasets that enable standardized benchmarks and (2) realistic datasets are both important for evaluating offline RL methods. However, D4RL already accounts for this.
> > >
> > > **1. D4RL already includes real data.** For “real human behavior”, the FrankaKitchen domain and Adroit domain consist of real human demonstrations collected via teleoperation, as mentioned in Section 5. This behavior is contained in the adroit-*-human tasks for the Adroit domain, and all FrankaKitchen tasks.
> > >
> > > In terms of including mathematical models of real behavior as you suggest, we utilize the “Intelligent driver model” [1] in our Flow tasks as outlined in Section 5. This is a well-cited and widely used model of human driving behavior within the operations research and traffic communities, keeping in line with our design principle of using well-vetted, realistic models and simulations. We labeled these tasks as flow-\*-controller, but perhaps it would be better to name them flow-\*-idm to make this fact more clear.
> > >
> > > [1] “Congested traffic states in empirical observations and microscopic simulations.” Treiber et. al, 2000
> > >
> > > **2. We do not claim that D4RL evaluates real-world applications.** While D4RL includes realistic data sources, such as data from humans, it is meant to be a mechanism for benchmarking offline RL algorithms, not for evaluating whether such algorithms can actually solve real-world applications. This is in line with previous benchmarks in the RL community (e.g., ALE, OpenAI gym, etc.). We disagree with the statement that "The authors are currently focusing their paper on real behavioral datasets" -- while D4RL includes real data, this is not the focus, and we do not claim that D4RL is anything but a synthetic benchmark and our design choices reflect this. However, if you believe that the current draft makes this claim, we would be happy to correct this issue and revise accordingly.
> > >
> > > > “On the other hand, the authors also don't generate good synthetic behaviors. Their behaviors are not reproducible..”
> > >
> > > We are not clear on why our behaviors are “not reproducible”, and why our synthetic datasets are not “good” - would it be possible to clarify on this point? For most of our datasets, we either include scripts for generating our datasets, or describe how our datasets were generated (e.g. in the case of our human datasets). We believe this should be sufficient for reproducing the majority of the datasets in our benchmark.
> > >
> > > > “In case of synthetic datasets, the authors should provide clean, simple datasets (even tabular), make the behaviors easy to reproduce, and make a notion of optimality clear.”
> > >
> > > We have included a Gridworld environment based on Minigrid (https://github.com/maximecb/gym-minigrid) for debugging purposes. These can be found in the code attached on our website (https://sites.google.com/view/d4rl-anonymous/), as the minigrid-fourrooms-random and minigrid-fourrooms tasks. Minigrid-fourrooms-random is collected via random actions, and minigrid-fourrooms uses the same collection method as the maze2d and antmaze environment via planning routes to randomly selected goals.
> > >
> > > We would be happy to make adjustments as per your request to improve D4RL, but your current criticisms do not provide a way of doing this. Your comment asks for: (a) real data (we already have this); (b) tabular domains (now added); (c) ease of reproducibility (this is already addressed -- all code is open-sourced and reproducible, but if there is any particular thing you would like us to do to improve reproducibility, please tell us). While we would be happy to improve D4RL to address your suggestions, currently the main issue you raise is already addressed (e.g., human data, mathematical models of real behavior).
> > >
> > > Lastly, we would conclude by noting that the RL community does have a standard for benchmarks. We believe that D4RL meets or exceeds this standard. While certainly there is more that could be done -- for example, developing higher fidelity simulators, tasks that cover other domains such as healthcare and education, tasks that cover additional properties such as casual structures, etc., we believe that D4RL already adds a lot in terms of allowing the community to standardize around an effective evaluation standard for offline RL methods. It is substantially better than the previous widely accepted standard, which was to use ad-hoc data generated from RL policies on MuJoCo gym locomotion tasks. If you have concerns that D4RL is a step in the wrong direction, please tell us what you think would be better, but so far we do not see a clear path for improvement from your comments.

---

### Official Review · AnonReviewer2 · 2020-10-19
**Concerns about accessibility for underrepresented groups**

**Rating:** 6
**Confidence:** 5

**Review:**

The paper proposes a standardized benchmark for offline RL research. The data collection is well motivated from real world scenarios covering many important design factors. I really appreciate that human demonstration and hand-crafted controllers are also included. The evaluation protocol and the API looks clear and easy to use. The benchmark of existing methods is thorough and provides many useful insights. I believe this work will have a high impact on the offline RL community. I can expect this benchmark will be used by many papers in the future and will function as the starting point for many offline RL research.

However, I notice that this dataset may not be accessible for underrepresented groups. I therefore vote to reject. As the authors note in the paper, each task consists of a dataset for training and a simulator for evaluation. In my understanding, half of the six tasks (Maze2D, AntMaze, Gym-mujoco) depend heavily on the MuJoCo simulator, which is a commercial software and is not free even for academic use. A personal MuJoCo license costs 500 USD per year. I am concerned that MuJoCo is not accessible for most underrepresented researchers. It is not clear when MuJoCo becomes a dominating benchmark for online RL research, though there are indeed free, open-sourced alternatives, e.g. PyBullet (https://github.com/bulletphysics/bullet3). In online RL, we need the simulator for training. One reason MuJoCo becomes popular may be because it's more stable and faster than PyBullet. However, in offline RL, a simulator is used only for evaluation not for training. So the high reliability of MuJoCo may no longer be so necessary. I therefore view offline RL as a good opportunity for the community to get rid of commercial simulation softwares, making RL research more accessible for underrepresented groups. If accepted, this paper will indeed greatly promote the use of MuJoCo given its potential high impact, making RL more privileged.

Overall, I really enjoy reading the paper and am glad to see a standardized benchmark for offline RL. I am happy to raise my score if the accessibility issue is addressed, e.g., by using PyBullet as the physical engine.

====================

(Nov 24) I appreciate the effort put into the Bullet reimplementation and therefore increase my score from 3 to 6.

---

> ### Comment · AnonReviewer4 · 2020-11-10
> **commercial software included**
>
> When writing my review I didn't realize that this benchmark requires commercial software (Mujoco). I fully agree with R2 that this is a very unfavorable (to put it mildly) prerequisite for a standardized benchmark.

---

> ### Public Comment · ~Edward_Grefenstette1 · 2020-11-12
> **Shockingly irresponsible reviewing, despite good intentions**
>
> I preface this comment by saying that I do not have any involvement in this paper, do not have knowledge of who its authors are, or any stake in its success. Furthermore, I actually agree with AnonReviewer2 that the use of MuJoCo is problematic, as while the maintainers offer a free personal student license, its terms are quite limited.
>
> However, literally punishing the authors of a paper for using this environment in their paper is a complete misuse of the reviewer's role and responsibilities. If the results of the paper stand and are scientifically valid and interesting, it is completely inappropriate to significantly reduce your score because you doesn't agree with terms of the experimental environment which the user doesn't have control over.
>
> If the reviewer will not change their score and assessment, the area chair should discard this review ~and seek to ensure the reviewer is not invited back to review for the conference.~ <- **edit:** fine, I recognise the reviewer is not acting in bad faith or maliciously, but I maintain my position regarding the review signal offered: if the paper makes a valuable contribution, it is fine to suggest the production of a version without the dependency on Mujoco, perhaps as follow up work. The beef the reviewer has is with the terms a third party is offering on the use of their library, not with the proposed scientific value of this submission.

---

> > ### Comment · AnonReviewer4 · 2020-11-12
> > **discussion culture**
> >
> > I don't know if I should take the comment "Shockingly irresponsible reviewing, despite good intentions" seriously or just see it as an emotional overreaction.
> > One can always have different opinions, whether e.g. the proposal of a standardized benchmark suite containing payment software should be published by ICLR or not. In fact, one task of the review process is to assemble these different opinions into a differentiated picture.
> > But to threaten the reviewer by demanding measures against the PERSON of the reviewer exceeds in my opinion a red line of the discussion culture.

---

> > > ### Public Comment · ~Edward_Grefenstette1 · 2020-11-12
> > > **Is this the place for this discussion?**
> > >
> > > On the other end of this review process are the authors. Perhaps this are junior, perhaps they are students. Why should they be the victims of a mode of discussion where the fate of their work hangs in the balance, when the criticism is one levelled at community practices rather than the scientific content of their paper?
> > >
> > > I am not threatening anyone. I am saying that the reviewer is behaving inappropriately by grounding their decision to reject in the cost of the license of the environment used to evaluate a scientific idea. The [ICLR code of ethics](https://iclr.cc/public/CodeOfEthics) calls upon authors and reviewers alike to uphold scientific standards and avoid harm, and rejecting work on a basis other than its scientific merit fails foul of the former criterion, and through the impact on the authors, the latter.
> > >
> > > I mean what next, do we take back Higgs' Nobel Prize because the LHC was very expensive to build, and not everyone got to use it for their own experiments? (Yes, I'm aware his contribution was not in the running of the actual experiments).

---

> > > > ### Comment · Area_Chair1 · 2020-11-12
> > > > **The license issue is understood**
> > > >
> > > > Thank you very much for the discussion.  Accessibility is an important factor to take into account particularly for a paper that proposes benchmarks but should not be the only point of discussion.  What are also important are the new ideas and knowledge presented by the paper.  AnonReviewer1 raises critical issues in this regard, and I encourage discussion there.  The license issue is understood, and I do not think any more discussion is needed on this point.  (Do not worry too much about the rating score.  We will take everything into account to make the final decision.)

---

> ### Public Comment · ~Rasool_Fakoor1 · 2020-11-12
> **unfair score**
>
> While I agree with you about the limitation of a paid license, I vehemently and respectfully disagree with you about your reasoning for rejection as this review is very unfair to the authors and it ignores all paper contributions altogether and rejects the paper just because it uses MuJoCo.  If we go with the same [rejection] logic, should all papers that use MuJoCo or have access to an unlimited compute be rejected too?
> As you know, MuJoCo is widely adopted in RL, and the majority of papers if not all that consider continuous control utilize MuJoCo. Hence that was(is) a natural choice to go with MuJoCo.
>
> Moreover, it is a very unreasonable request to ask authors to create the same benchmark for PyBullet during rebuttal. If we ignore all compute costs and human efforts, it needs months to create such a benchmark for PyBullet, i.e. 40+ environments * 3 seeds * 10+ baselines ~ 1200 different runs.
>
> Lastly, this kind of logic for paper rejection will only discourage researchers from investing in building such benchmarks which ultimately puts a negative effect on doing research in these areas.
>
> I urge you to judge this paper based on its merits, I don't suggest acceptance or rejection of this paper,  I am just asking to judge this paper based on its merits.
>
>
> PS: I am not an author or have any involvement in this paper.

---

> ### Comment · AnonReviewer2 · 2020-11-12
> **Some clarification**
>
> I want to make it clear that this paper is not the case where a new algorithm is proposed and Mujoco is used as an evaluation benchmark or Mujoco is used to better understand existing algorithms. The main contribution of this paper is to propose a standardized benchmark. And in my understanding, even if the proposed benchmark is publicly released, Mujoco is still necessary for practitioners to use most of the proposed benchmark.  In light of this, I do not think this contribution adheres well to the following points quoted from the ICLR Code of Ethics.
>
> >> Researchers should foster fair participation of all people—in their research, at the conference and generally—including those of underrepresented groups.
>
> >> The use of information and technology may cause new, or enhance existing, inequities. Technologies and practices should be as inclusive and accessible as possible and researchers should take action to avoid creating systems or technologies that disenfranchise or oppress people.
>
> If there is other contribution beyond the proposal of this new benchmark, I would like to evaluate the document based on other contributions. However, I do not find other significant contributions on the current version of the document.

---

> ### Comment · AnonReviewer1 · 2020-11-14
> **Accessibility Problem**
>
> While I tend to agree with you, I feel that following this line of thought, 99% of research papers in ML may be desk-rejected.
> In order to be able to reproduce, use, or advance much of the concurrent ML work, one needs access to expensive CUDA enabled GPUs and/or large amount of CPU power. This inherently biases research in ML toward rich companies with access to such compute. While I see your case, I do not fully understand why this would not fall into the same category for rejection.

---

> ### Author Response · Authors · 2020-11-16
> **Author's response**
>
> Thank you for your comments, and raising awareness on this important issue. We are glad to hear that you think this benchmark has potential to have a high impact on the RL community and will be used by many papers to come.
>
> We agree that using free, open-source simulators would benefit the community and the introduction of a new benchmark is a great place to make that shift in the community. We have begun implementing a PyBullet version of our tasks, and will add an update here on the progress before the rebuttal deadline.
>
> Nevertheless, MuJoCo is already used by the community including many RL benchmark papers such as RLLAB [1], RLUnplugged [2], Metaworld [3], and Gym [4]. However, adding PyBullet versions to the current tasks seems like the best path forward. We also emphasize that several of our domains, such as CARLA and Flow, use simulators that do not require a paid license.
>
> We note that the authors are not affiliated with the company that sells MuJoCo. MuJoCo does offer free licenses on its website for personal use, for projects not “part of employment” and not already receiving financial support.
>
> [1] “Benchmarking Deep Reinforcement Learning for Continuous Control” Duan et. al. ICML 2016
>
> [2] “RL Unplugged: Benchmarks for Offline Reinforcement Learning” Gulcehre et. al. NeurIPS 2020
>
> [3] “Meta-World: A Benchmark and Evaluation for Multi-Task and Meta Reinforcement Learning“ Yu et. al. CoRL 2019
>
> [4] “OpenAI Gym” Brockman et. al. 2016

---

> > ### Author Response · Authors · 2020-11-22
> > **Progress Update on Bullet Re-implementation**
> >
> > We are pleased to report that we are nearing completion for reimplementing the Gym-MuJoCo and Maze2D tasks using the Bullet simulator. The dataset URLs and environment code are included in the code on our website at https://sites.google.com/view/d4rl-anonymous/. We will be continuing to work on integrating this into the rest of the library, and are planning out how to reimplement AntMaze next as well.

---

> > > ### Comment · AnonReviewer2 · 2020-11-24
> > > **Response about PyBullet**
> > >
> > > I appreciate the effort put into the Bullet reimplementation, which I believe is an important contribution of this paper. Given the current process, I feel confident that all the reimplementation will be done in the camera-ready copy. I have increased my score accordingly.

---

### Official Review · AnonReviewer3 · 2020-10-27
**Review from reviewer 3**

**Rating:** 6
**Confidence:** 5

**Review:**

D4RL: Datasets for Deep Data-Driven Reinforcement Learning
review:

summarization:

In this paper, the authors consider the problems of offline reinforcement learning problems,
 and has a focus of dataset and shareable code base.
While no novel algorithms are proposed in this project,
a systematic evaluation of existing algorithms (offline RL algorithms) is proposed.

Pros:
1. Offline RL is a hot topic this year,
with a lot of papers exploring efficient and robust ways to utilize offline collected data.
This paper, while using simulated data,
provides a general platform to benchmark these algorithms.

2. The project provides a comprehensive evaluation and discussion on existing considerations in offline RL.
It provides several interesting directions in offline RL.

3. The paper is well-written.
It is very clear what the purpose of the project is,
and it is very clear why the authors make the dataset in the way they did.

Cons:

1. The dataset is mostly simulated.
While I understand the difficulty of collection real-data,
it does raise some concerns that a simulated dataset can be generated by researchers themselves.

Summary:
While the dataset is simulated,
I do think there’s value in the dataset and the shared code-base to facilitate recent progress in offline RL research.

---

> ### Author Response · Authors · 2020-11-16
> **Clarifications on simulation**
>
> > “The dataset is mostly simulated. While I understand the difficulty of collection real-data, it does raise some concerns that a simulated dataset can be generated by researchers themselves.”
>
> Thank you for your comments. We made a very conscious choice in the design of the benchmark to keep everything in simulation, and select domains for which realistic, accurate simulations exist and have been vetted by the research community. We would be open to suggestions which allow both accurate evaluation and a convincing degree of realism in order to improve the benchmark.
>
> We would also like some additional clarification on what concerns you have with using a dataset generated by a simulator, and we can add this to the discussion in the main text.

---

### Official Review · AnonReviewer4 · 2020-10-28
**Premature or high time?**

**Rating:** 6
**Confidence:** 4

**Review:**

Summary:
In this paper a test suite of data sets and corresponding benchmarks for offline reinforcement learning is introduced.
Several existing RL benchmarks are used, the results of several algorithms are presented.
The authors claim that the benchmarks were specifically designed for the offline setting and are guided by the key properties of datasets in real-world applications of offline RL.

Strong points:
The present paper has already been cited and the benchmarks suite has already been used by other publications. Obviously there is a need for offline RL test suites.

Weak points:
The authors' claim that such a benchmark for offline RL should "be composed of tasks that reflect challenges in real-world applications of data-driven RL" is only partially met by the paper in its present form. The area of robotics, with deterministic dynamics, is comparatively well represented, but there is no real, industrial application. In particular it seems that so far no benchmark has been included that has the ambition to have the characteristics and complexity of a real application.

Recommendation:
On the one hand, the really realistic benchmarks are missing, so that a publication seems premature. On the other hand, the current status is already used by the research community, since there seems to be no test suite for offline RL apart from "RL unplugged: Benchmarks for offline reinforcement learning". I therefore recommend to accept the paper.

Questions:
To what extent are the current benchmarks stochastic?
Are there bi- or multimodal transition probabilities?

Additional feedback with the aim to improve the paper:
In Table 2 and Table 3 average results are reported over only 3 random seeds. This seems to me to be clearly too little, especially since the policy performance of Q-function based algorithms often fluctuates strongly. Since no uncertainties, e.g. in the form of standard error, are given, the reliability of the results cannot be assessed.
The way policies are selected before they are tested should be described more clearly. My impression was that in each case the policy is used that results for a considered algorithm and random seed after 500K training iterations or gradient steps. Since different algorithms require different computational efforts this approach does not seem to be in the sense of a real-world application. In most cases there should be the willingness to use much more computational effort for especially good policies. According to the motto: computing power is cheap, data is expensive.

I like the formulation “Effective offline RL algorithms must handle […] data collected via processes that may not be representable by the chosen policy class.“ This expresses the, in my opinion, correct view of the real situation well, while the assumption that there is a "behavior policy" that generated the data is not true in general. It may have been different people at different times who performed the actions while the data set was recorded.

Please check the bibliography for accidental lower case, like „markov“, „adobeindoornav“


----------------------------------
(Dec 3) Taking into account the other reviews, the authors' responses and the changes made by the authors, as well as the extensive and controversial discussion, I rate the paper still with a score of 6.

---

> ### Author Response · Authors · 2020-11-16
> **Rebuttal for reviewer 4**
>
> Thank you for your feedback. We believe your main concern is in the real-world ramifications of the tasks used in the benchmark, to which we respond to below, along with several other clarifications. Please let us know if you have any additional questions, and if this addresses your concerns.
>
> > “The authors' claim that such a benchmark for offline RL should "be composed of tasks that reflect challenges in real-world applications of data-driven RL" is only partially met by the paper in its present form. The area of robotics, with deterministic dynamics, is comparatively well represented, but there is no real, industrial application… on the one hand, the really realistic benchmarks are missing”
>
> We agree that incorporating tasks with real-life implications is important for a benchmark, and therefore we leveraged some of the most realistic simulated domains which are widely available for research use. For example, the Adroit and FrankaKitchen domains are models of Shadow Hand and Franka robots, respectively, and leverage real human demonstrations collected via motion capture. Likewise, CARLA is a photorealistic simulator used in the autonomous driving community (e.g. [1]) and Flow is a traffic simulator used in the operations research/transportation community (e.g. [2]). Robotic manipulation, autonomous driving, and traffic control are all domains with real, challenging applications and significant implications for everyday life.
>
> We considered other domains with real datasets, such as healthcare (MIMIC-III) or recommender systems (various, across industry), but for the purposes of constructing a benchmark, we do not have a reliable method for evaluating the performance of algorithms on these tasks aside from costly real-world evaluation.
>
> [1] “End-to-end driving via conditional imitation learning.” Codevilla et. al. ICRA 2018
>
> [2] “Stabilizing traffic with autonomous vehicles” Wu et. al. ICRA 2018
>
> > “To what extent are the current benchmarks stochastic? Are there bi- or multimodal transition probabilities?”
>
> The current benchmarks are stochastic in the initial state (i.e. in the navigation environments the starting location is randomized), and through partial observability in the case of CARLA. Stochasticity is however, a property that is not explored in-depth compared to the other properties we outlined in Section 4. We added additional discussion of this point in Section 7.
>
> > “… average results are reported over only 3 random seeds. This seems to me to be clearly too little, especially since the policy performance of Q-function based algorithms often fluctuates strongly. … In most cases there should be the willingness to use much more computational effort for especially good policies. According to the motto: computing power is cheap, data is expensive.”
>
> The cost for evaluating this benchmark was already quite expensive due to its scope, with 11 algorithms evaluated and 42 tasks (for a total of 462 evaluations), including several which require GPU access. We will evaluate additional seeds, but this may not be complete during the rebuttal period.

---

> > ### Comment · AnonReviewer4 · 2020-11-19
> > **Attempt of clarification**
> >
> > Thanks for the feedback.
> > Before I address the overall feedback, I would like to point out that the last part of the feedback, namely  “… average results are reported over only 3 random seeds. This seems to me to be clearly too little, especially since the policy performance of Q-function based algorithms often fluctuates strongly. … In most cases there should be the willingness to use much more computational effort for especially good policies. According to the motto: computing power is cheap, data is expensive.”, connects two aspects that I intended to be considered isolated. There seems to have been a misunderstanding.
> >
> > The request for more than 3 random seeds, in combination with the request for the specification of the standard error, is intended to make the results reliable, otherwise the entire results presented in tables 2 and 3 cannot be used to provide a reference for the performance, nor to compare the performance of the different methods.
> >
> > Independently from this, the critique has to be seen, that all algorithms are being sheared over the same comb regarding the number of update steps, because in practice there is a willingness in many applications to use sufficient computing time. What is meant is that each algorithm should get as much computing time as necessary, because the challenge in offline RL is primarily to cope with little data (or with little variation in the data) and less with little computing time.

---

> > ### Comment · AnonReviewer4 · 2020-11-23
> > **Concerning "Rebuttal for reviewer 4"**
> >
> > I appreciate the feedback from the authors "Rebuttal for reviewer 4" and the additions already included in the text. Nevertheless, my assessment expressed in the review "Premature or high time?" has not changed due to this feedback, as the composition of the benchmarks as well as my assessment of the relevance and balance of the benchmarks remains unchanged.

---

### Comment · Area_Chair1 · 2020-11-22
**Reviewers, please read responses**

Dear reviewers,

Thank you very much for your reviews.  The authors have given detailed responses to your concerns.  Please give comments regarding whether the responses have resolved your concerns or you still see major issues.

Reviewer 1,

How did the responses from the authors change or not change your opinions?  You had the most negative score, but it seems the authors have provided strong responses.

---

> ### Comment · AnonReviewer1 · 2020-11-22
> **re: my response**
>
> I have responded to the authors' rebuttal, and will be happy to keep a live discussion on the matter.
>
> Shortly, the main problem I see is that these offline benchmarks are synthetic and do not represent real world data.
>
> What is missing from this work is something that would convince me that getting a high score on these dataset has some meaning. How does solving the offline RL tasks reflect to the real world? Why are these benchmarks interesting? Aside from the authors' claims, there is no evidence their datasets represent anything from the real world. A fundamental problem is the synthetic behavior that generated the data, which is not representative of anything realistic (unless proven otherwise).

---

> > ### Comment · AnonReviewer4 · 2020-11-24
> > **How does solving the offline RL tasks reflect to the real world?**
> >
> > I think, with the question „How does solving the offline RL tasks reflect to the real world?“ Reviewer1 has raised a very central point concerning the usefulness of benchmarks.

---

### Author Response · Authors · 2020-11-24
**Summary of changes**

We appreciate the constructive feedback from the reviewers. In response, we:
- are nearly complete reimplementing the Gym-MuJoCo and Maze2D tasks using the PyBullet simulator. Progress so far is available at https://sites.google.com/view/d4rl-anonymous/.
- are in the process of running more seeds (10 seeds) to provide stronger statistical significance.
- have revised the text to clarify that D4RL includes real behavior data from humans and includes tasks that use models of realistic systems that are accepted as such by domain experts (Flow, Franka & Adroit, Carla).

The reviewers have pointed out areas for improvement and discussion. However, no benchmark is perfect and compared to the closest similar work, D4RL makes significant contributions in important areas:
- includes real behavior data and synthetic behavior data that incorporates narrow data distributions, multitask data, and non-representable policies. In prior work, the predominant method for evaluating offline RL algorithms was to generate data from online RL agents on Atari or Gym tasks. As we show in our empirical evaluation, this does not exercise important dimensions of the problem.
- is reproducible: all our code and data is open-source, we include scripts for generating our datasets or describe how our datasets were generated (e.g. in the case of our human datasets), and can be used by anyone (PyBullet, CARLA, Flow do not require a paid license, and MuJoCo has a free student license). Previous work either does not open source the datasets and/or the behavior policies used to generate the datasets.
- provides a comprehensive evaluation of 11 RL methods. Previously, no such extensive evaluation on a common dataset had been done in offline RL.

---

### Comment · Area_Chair1 · 2020-11-24
**More comments on behavior policies?**

I thank all participants for very active discussion.  I would like to summarize a few key points of discussion, including my own observations, so far by first drawing attention to the distinction between synthetic/simulated/real environments and synthetic/real behavior policies (https://openreview.net/forum?id=px0-N3_KjA&noteId=L0o4qw789AQ).

Regarding the environments, there were some concerns about whether the environments are sufficiently accessible or realistic.  For accessibility, the authors added reimplementation with PyBullet, which I assume has solved the issues associated with MuJoCo.  There are some disagreement regarding whether the environments are sufficiently realistic or not.

On one hand, I understand the criticisms and disappointment about the use of simulators, because the needs of offline RL come from exactly those situations where simulators are unavailable.  One would expect to see the datasets generated from the environments that have not been used in online RL research.

On the other hand, simulators are essential to evaluate the policies learned with batch RL.  Although there exist methods of offline (off policy) policy evaluation, their reliability is much lower than the evaluation with simulators.

Overall, although there might be some room of improvement, it appears that the proposed datasets have good coverage of environments if we accept that simulators must exist for offline RL datasets (please speak up if you disagree).

There are also some disagreement regarding the behavior policies used to generate the data.  In my understanding, the choice of behavior policies is critically important in offline RL datasets.  A key role of an offline RL dataset is to present common behavior data (or equivalently behavior policies) that the community should use.  So, those behavior policies must be better than arbitrary behavior policies in some sense.  Real or human behavior policies would be good in the sense that they are realistic.  There might also exist good synthetic behavior policies, which are particularly suitable for evaluating offline RL methods.

What could be made clear is whether there are novel contributions in the way behavior policies are selected, whether the choice of the behavior policies is the best with the current state of the art, or whether the choice of behavior policies is unsatisfactory.

---

> ### Comment · AnonReviewer4 · 2020-11-24
> **Some comments**
>
> „For accessibility, the authors added reimplementation with PyBullet, which I assume has solved the issues associated with MuJoCo.“
> In my opinion, this is an impressive achievement of the authors. It improves the benchmark collection and avoids possible conflicts with the ICLR CoE.
>
>
> “…it appears that the proposed datasets have good coverage of environments”
> On the one hand I would like to respectfully disagree, on the other hand I would like to point out that this comment seems to be the personal view of Area Chair1 and cannot serve as summary of the opinions of all reviewers. In my opinion, Reviewer1 (https://openreview.net/forum?id=px0-N3_KjA&noteId=_H97HrjdmeR)
> and Reviewer4 (https://openreview.net/forum?id=px0-N3_KjA&noteId=iBlZrFsxYPE, see Weak points) have expressed the opposite opinion.
>
>
> “if we accept that simulators must exist for offline RL datasets (please speak up if you disagree).”
> Here I fully agree, because a pure real-world data set is simply not sufficient to measure the performance of the created policy exactly. Therefore a simulation (or the real system) is needed.
>
>
> „Real or human behavior policies”
> Please excuse me if it sounds pedantic, but the term "behavior policies" is used imprecisely in my opinion, because in real data sets there often exists no "behavior policy" that produced the data, as the authors very nicely phrase it:
> “Effective offline RL algorithms must handle […] data collected via processes that may not be representable by the chosen policy class.“ This expresses the, in my opinion, correct view of the real situation well, while the assumption that there is a "behavior policy" that generated the data is not true in general. It may have been different people at different times who performed the actions while the data set was recorded.

---

> ### Author Response · Authors · 2020-11-24
> **On behavior policies.**
>
> > “What could be made clear is whether there are novel contributions in the way behavior policies are selected, whether the choice of the behavior policies is the best with the current state of the art, or whether the choice of behavior policies is unsatisfactory.”
>
> We can attempt clarify on the novelty of the way behavior policies are selected, and other reviewers can comment if they feel this is not accurate.
>
> There have been several behavior policies proposed in prior work which we incorporate, which include using data contained in the replay buffer, and using data collected from RL-trained policies of varying performance. As prior papers typically evaluate on a subset of these problems, we decided to include all of these data collection methods in the benchmark, and evaluate each one. These are primarily incorporated in the Gym-MuJoCo domain.
>
> In terms of datasets that have not believe have been proposed in prior work, we believe the following data collection methods have not been widely incorporated in offline RL:
> - **Human data.** The Adroit and FrankaKitchen domains contain datasets acquired from human demonstrations. Utilizing human data is likely an important step for making progress towards real applications of offline RL.
> - **Hand-designed controllers.** We also include data collected from fixed, hand-designed controllers which are importantly not RL-trained policies (navigation domains, Flow). As discussed in the text, this can potentially stress-test the ability of algorithms to handle behavior policies outside of the function class.
> - **Behavior policies not aimed at solving the task of interest** (i.e. task-unrelated behavior). On the FrankaKitchen domain, the behavior policy consists of behaviors that attempt to perform auxiliary tasks such as opening cabinets, the microwave, moving the kettle, etc. however not all of these are actually needed to, or relevant towards optimizing the reward function. This is indicative of real-world applications where we wish to utilize general-purpose task-agnostic data to optimize a given reward function.
> - **Passively logged data**. On our navigation domains (Maze2D, AntMaze, CARLA), we include data observed from random, undirected navigation, which are retroactively labeled with a task reward. We believe passive logging will be similar to how many large datasets will be collected in realistic applications.
> - **Mixture datasets**. We include datasets (on the Gym domain) which consist of mixtures of behavior policies. This can potentially stress-test the ability of algorithms to handle multi-model state, action and trajectory distributions.

---

### Decision · Program_Chairs · 2021-01-07
**Final Decision**

**Decision:**

Reject

**Comment:**

This paper proposes benchmark tasks for offline reinforcement learning.  The paper has major strength and weakness, and it has resulted in very active discussion among reviewers, authors, and other participants.

The major strength includes the following:
- The proposed benchmark is already heavily used in the community
- Offline reinforcement learning is very important to solve reinforcement learning tasks in the real world
- The paper covers a range of tasks and provides through evaluation of existing methods to be used as baselines

The major weakness is that it is not sufficiently convincing that the methods that perform well in the proposed benchmark tasks will perform well in the offline reinforcement learning tasks in the real world.

This is partly due to the nature of the benchmark tasks of offline reinforcement learning, which require simulators to evaluate the policies learned with offline reinforcement learning.  This means that one cannot simply collect datasets from real world tasks and provide them as benchmark datasets.

Although one cannot do much about simulators, benchmark tasks for offline reinforcement learning still have many design choices.  In particular, how should the datasets in the benchmark be collected (i.e., behavior policies)?

While the datasets in the proposed benchmark are collected with various behavior policies including humans, it is not necessarily convincing that the resulting benchmark tasks are good for the purpose of evaluating offline reinforcement learning to be used in the real world.

In addition to the suggestions given by the reviewers, a possible direction to improve the paper is to focus on the choice of behavior policies used to generate the datasets in the proposed benchmark.  One might then be able to provide some convincing arguments as to why performing well in the benchmark might imply good performance in the real world by relating it to the choice of behavior policies.

---

> ### Comment · ~Sergey_Levine2 · 2021-02-03
> **Clarification**
>
> Dear Area Chair,
>
> I appreciate your evaluation of the work, though I respectfully disagree with your assessment.
>
> However, it is important for us to serve the community effectively by improving the benchmark, since it is now the most widely used mechanism for evaluating offline RL methods. May I ask for clarification for what you consider to be a better choice of behavior policies used to generate the datasets? Since this appears to be the only issue with the work that you bring up, it seems reasonable for me to conclude that it must be quite important.
>
> The question of how we can effectively evaluate offline RL algorithms is important to me, and I would like to find a mechanism that works well for the entire community.
>
> If you would like, you are also welcome to contact me privately by email, or else reply to this comment.
>
> Thank you,
>
> Sergey Levine

---

> > ### Comment · Area_Chair1 · 2021-02-03
> > **Clarification**
> >
> > I would like to clarify that the main point of the comment was that "[t]he major weakness is that it is not sufficiently convincing that the methods that perform well in the proposed benchmark tasks will perform well in the offline reinforcement learning tasks in the real world."  I suggested the choice of behavior polices as a potential direction to convince a wider range of audience that "the resulting benchmark tasks are good for the purpose of evaluating offline reinforcement learning to be used in the real world," but there might be other directions.  I am not in the position of proposing better behavior policies.  You might have selected the best behavior policies, and it might just be that the paper does not provide sufficient evidence that those are a good choice.
> >
> > Finally, I am confident that I did my best to lead the discussion to evaluate the paper from various perspectives, that all of the reviewers have actively participated in the discussion and sometimes changed their opinions based on the discussion, and that I have made a fair recommendation taking into account everything in the reviews and discussion with extra care for this particular paper.  I hope you also understand that there is a limited capacity for the conference, and some of good borderline papers were rejected.